# circTP63 functions as a ceRNA to promote lung squamous cell carcinoma progression by upregulating FOXM1

Zhuoan Cheng[1,6], Chengtao Yu[1,6], Shaohua Cui[2,6], Hui Wang[3], Haojie Jin[3], Cun Wang[3], Botai Li[1], Meilin Qin[3], Chen Yang[4], Jia He[3], Qiaozhu Zuo[3], Siying Wang[3], Jun Liu [2], Weidong Ye[5], Yuanyuan Lv[3], Fangyu Zhao[3], Ming Yao[3], Liyan Jiang[2] & Wenxin Qin [1,3]

Circular RNAs (circRNAs) are identified as vital regulators in a variety of cancers. However, the role of circRNA in lung squamous cell carcinoma (LUSC) remains largely unknown. Herein, we explore the expression profiles of circRNA and mRNA in 5 paired samples of LUSC. By analyzing the co-expression network of differentially expressed circRNAs and dysregulated mRNAs, we identify that a cell cycle-related circRNA, circTP63, is upregulated in LUSC tissues and its upregulation is correlated with larger tumor size and higher TNM stage in LUSC patients. Elevated circTP63 promotes cell proliferation both in vitro and in vivo. Mechanistically, circTP63 shares miRNA response elements with FOXM1. circTP63 competitively binds to miR-873-3p and prevents miR-873-3p to decrease the level of FOXM1, which upregulates CENPA and CENPB, and finally facilitates cell cycle progression.

[1] State Key Laboratory of Oncogenes and Related Genes, Shanghai Cancer Institute, Renji Hospital, Shanghai Jiao Tong University School of Biomedical Engineering, 200032 Shanghai, China. [2] Department of Respiratory Medicine, Shanghai Chest Hospital, Shanghai Jiao Tong University, 200030 Shanghai, China. [3] Shanghai Cancer Institute, Renji Hospital, Shanghai Jiao Tong University School of Medicine, 200032 Shanghai, China. [4] Shanghai Medical College of Fudan University, 200032 Shanghai, China. [5] Department of General Surgery, Shanghai Jiao Tong University Affiliated Sixth People's Hospital, 200233 Shanghai, China. [6] These authors contributed equally: Zhuoan Cheng, Chengtao Yu, Shaohua Cui. Correspondence and requests for materials should be addressed to L.J. (email: jiang_liyan2000@126.com) or to W.Q. (email: wxqin@sjtu.edu.cn)

Lung cancer is the most common incident cancer and the leading cause of cancer death worldwide[1]. In China, there were ~0.73 million newly diagnosed and 0.61 million death due to lung cancer in 2015[2]. Overall, 80–85% of all human lung cancers are non-small cell lung cancer (NSCLC), of which lung squamous cell carcinoma (LUSC) and lung adenocarcinoma (LUAD) are the major subtypes. Recently, the development of targeted drugs for specific gene mutations has greatly improved the therapy of advanced LUAD patients. In contrast, only a small proportion of LUSC harbor driver gene mutations, leading to a 5-year survival rate <5% due to compromised efficacy of platinum-based chemotherapy for LUSC[3]. Therefore, it is critical to further address the molecular mechanisms underlying the development and progression of LUSC for the development of more effective treatment options.

Lung cancer results from multiple complex combinations of morphological, molecular, and genetic alterations[4]. Noncoding RNAs are transcribed from a large proportion of human genome and regulate gene expression[5]. Recently, circular RNAs (circRNAs) as a type of regulatory RNAs have attracted great research interest. circRNAs are characterized by covalently closed loop structures with neither 5′–3′ polarity nor a poly-adenylated tail[6]. They are abundant, conserved stable and tissue or developmental-stage specific[7,8]. With accumulated knowledge of characteristics and functions of circRNAs, it has been described that circRNAs play important roles in human diseases, such as atherosclerotic vascular disease[9], neurological disorders[10], heart disease[11], and cancer[12–14]. To date, circRNAs functioning as the competing endogenous RNA (ceRNA) have been most widely reported in various types of cancer. For example, multiple cancer-related CDR1as[15–18], hepatocellular carcinoma suppressor cSMARCA5[19] and circMTO1[20], and oncogenic circCCDC66 in colon cancer progression[21], act as miRNA sponges to involve in cancer development. Besides the ceRNA mechanism, circRNAs can interact with RNA-binding proteins to regulate gene expressions[22]and some of them can encode functional proteins[23,24]. Moreover, circRNAs have potential to be biomarkers for disease diagnosis[25,26].

In lung cancer, a group of circRNAs have been found to be significantly dysregulated[27,28] and several LUAD-related circRNAs are identified. For example, circPRKCI and Fusion-circEA1 (generated by the EMF4/ALK1 fusion gene) may act as oncogenic circRNAs[29,30]. circRNA-ITCH may serve as a tumor suppressor by upregulating ITCH expression[31]. Although circRNAs has been identified to be crucial for LUAD progression, the roles of circRNAs in LUSC are largely unknown. Xu and colleagues briefly investigate the expression profile of circRNAs in three LUSC and matched nontumorous tissues by an array analysis only containing probes for circRNA[32]. However, roles and mechanisms of circRNAs in LUSC have not been explored comprehensively.

In this study, we investigate the expression profiling of circRNA and mRNA in five LUSC and paired adjacent tissues through a microarray containing probes for circRNAs and mRNAs. Then, a significant upregulated circRNA, hsa_circ_0068515, designated as circTP63, is initially identified. circTP63 is correlated with larger tumor size and higher TNM stage in LUSC patients and promotes cell proliferation by functioning as a ceRNA to upregulate FOXM1. Our results indicate that circTP63 exerts oncogenic potential and it may be a candidate in diagnosis and treatment of LUSC.

## Results

### circTP63 is upregulated in LUSC.
We simultaneously analyzed the expression profiles of circRNA and mRNA in five paired samples of LUSC and matched nontumorous tissues by SBC Human ceRNA Array, which contains 88,371 circRNA probes, 77,103 lncRNA probes, and 18,853 mRNA probes (GEO Submisstion: GSE126533) (www.ncbi.nlm.nih.gov/geo). A total of 7081 dysregulated circRNAs were identified in LUSC tissues, of which 3157 circRNAs were upregulated and 3924 circRNAs were downregulated (step 1 in Fig. 1a and Supplementary Fig. 1a, b). In addition, 2832 differentially expressed mRNAs were also identified, with 979 mRNAs upregulated and 1853 mRNAs downregulated (step 1 in Fig. 1a and Supplementary Fig. 1c, d). To explore crucial circRNAs that involved in LUSC, we did co-expression network analysis between the top 100 mostly changed circRNAs (step 2 in Fig. 1a, b and Supplementary Table 1) and 109 dysregulated genes in cell cycle which was the main pathway revealed by KEGG pathway analysis (Fig. 1c). The network implied that 6 circRNAs and 79 mRNAs might involve in cell cycle regulation (step 3 in Fig. 1a and Supplementary Fig. 2a). To check the resistance of the six circRNAs to RNase R digestion, circRNA candidates were analyzed by reverse transcription PCR (RT-PCR) after RNase R treatment. The levels of linear isoform were used to illustrate the efficacy of RNase R treatment. Results showed that only hsa_circ_0068515 (termed circTP63) was more resistant to RNase R treatment compared to TP63 linear isoform (step 4 in Fig. 1a and d). hsa_circ_0026398 and hsa_circ_0074026 could not be detected due to their very low abundance in lung cancer cells. For hsa_circ_0026443, we tried different primers to amplify this circRNA, but all of these primers caused nonspecific amplification. hsa_circ_0026414 and hsa_circ_0019089 were sensitive to RNase R, suggesting that some identified circRNAs may be false positives. In addition, we selected other 10 circRNAs (five upregulated and five downregulated circRNAs) from the top 20 dysregulated circRNAs to verify the microarray results. Sanger sequencing confirmed their back-spiced junctions (Supplementary Fig. 1e). qRT-PCR analysis showed that the expression of these 10 circRNAs was consistent with the result of microarray (Supplementary Fig. 1f).

Notably, circTP63 was significantly upregulated according to the probe signal calculation of the microarray data (Fig. 1e). Next, we confirmed the upregulated expression of circTP63 in another 35 paired LUSC samples by quantitative reverse transcription PCR (qRT-PCR), and found that the expression of circTP63 was significantly higher in 65.7% (23 of 35) of LUSC tissues (Fig. 1f). In addition, the expression of TP63 in LUSC tissues was significantly higher compared to corresponding adjacent nontumorous tissues (Supplementary Fig. 2b). The expression of circTP63 was positively correlated with TP63 expression ($R = 0.719$, $p < 0.0001$) (Supplementary Fig. 2c), suggesting that higher expression of circTP63 was associated with higher expression of TP63. Importantly, increased circTP63 in LUSC tissues was significantly correlated with larger tumor size and higher TNM stage in LUSC patients (Table 1). We analyzed the correlation between circTP63 expression and prognosis of LUSC patients in this study. The preliminary Kaplan–Meier analysis showed that patients with higher level of circTP63 were more likely to be poor overall survivals (OS), although p-value was not significant ($p = 0.2930$, Supplementary Fig. 2d). Taken together, these results suggest that circTP63 upregulation is common in LUSC and its regulation is correlated with later clinical stage.

### Characteristics of circTP63 in LUSC cells.
We next assessed the exon structure of circTP63, which derived from exon 10 to 11 of TP63 gene with a length of 295 nt. The back-spliced junction of circTP63 was amplified using divergent primers and confirmed by Sanger sequencing (Fig. 2a). The sequence is consistent with circBase database annotation (http://www.circbase.org/). PCR

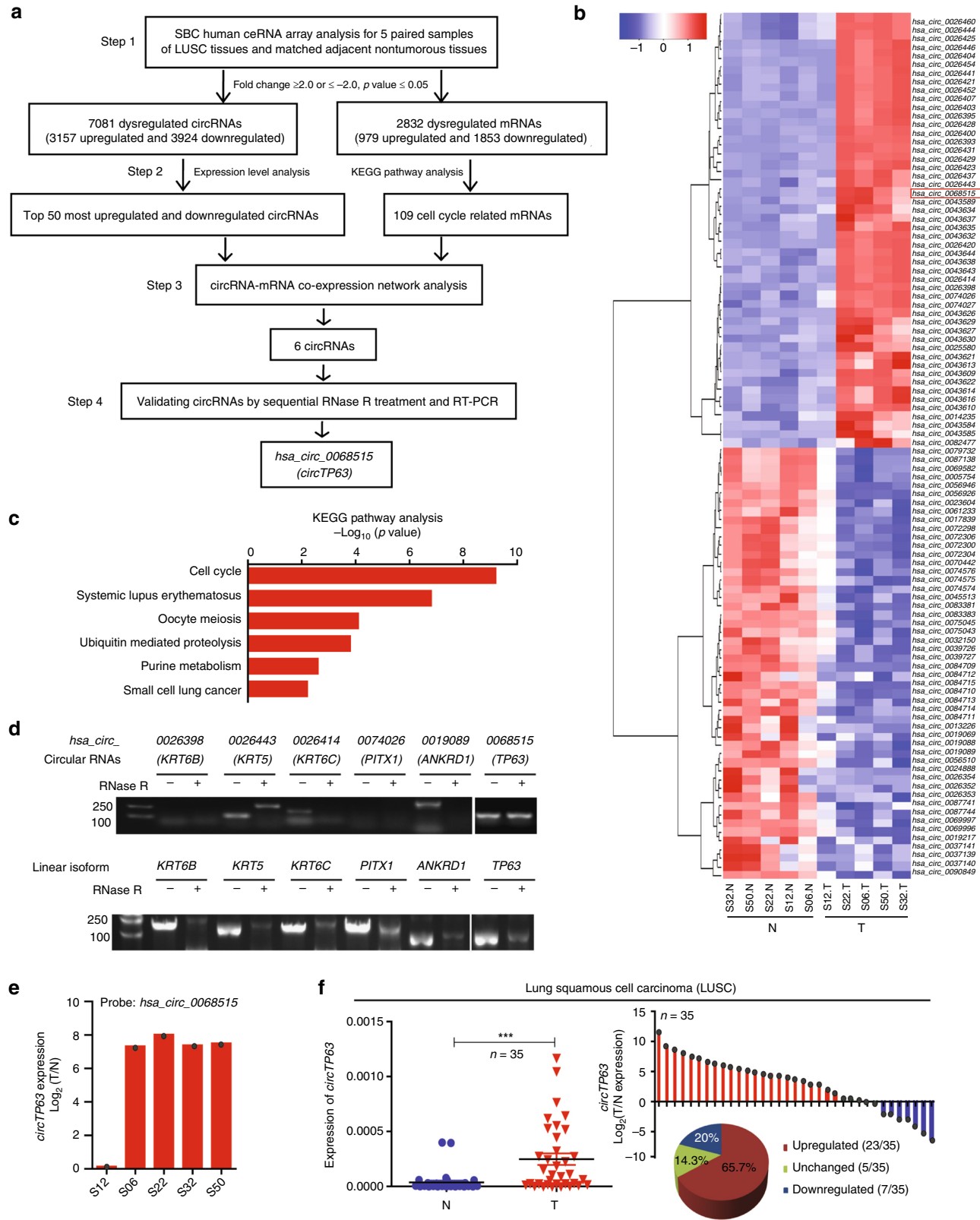

analysis for reverse-transcribed RNA (cDNA) and genomic DNA (gDNA) showed that divergent primers could amplify products from cDNA but not from gDNA (Fig. 2b). Northern blot analysis was performed to confirm that *circTP63* could be observed at 295nt with a probe targeted the back-spliced junction (Fig. 2c). Analysis for stability of *circTP63* and *TP63* in H1703 cells treated with Actinomycin D, an inhibitor of transcription, revealed that

the half-life of *circTP63* transcript exceeded 24 h, with more stable than *TP63* (Fig. 2d). To observe cellular localization of *circTP63*, we conducted qRT-PCR analysis for nuclear and cytoplasmic *circTP63* RNA. Results showed that *circTP63* transcript preferentially located in the cytoplasm (Fig. 2e). Furthermore, we tested the endogenous *circTP63* expression in six LUSC cell lines and two human normal lung cell lines. We found that *circTP63*

**Fig. 1** circRNA expression profiling reveals that *circTP63* is upregulated in LUSC. **a** The flowchart delineates the steps for identifying and validating circRNAs in LUSC. **b** A heatmap shows the top 50 most upregulated and top 50 most downregulated circRNAs in five paired samples of tumorous tissues (T) and corresponding adjacent nontumorous tissues (N) from patients with LUSC by SBC Human ceRNA Array analysis. **c** KEGG pathway analysis for the 2832 dysregulated mRNAs. **d** Validation of six circRNAs by RNase R treatment and reverse transcription PCR (RT-PCR) analysis. **e** Fold change of *circTP63* in the five paired samples of LUSC for SBC Human ceRNA Array analysis. **f** Left: expression levels of *circTP63* in additional 35 paired samples of LUSC were determined by quantitative reverse transcription PCR (qRT-PCR). *β-actin* was used as a loading control. Right: histogram and pie chart of the proportions of LUSC samples in which *circTP63* expression was upregulated (23/35, 65.7%, red), downregulated (7/35, 20%, blue), or no change (5/35, 14.3%, green). Log2(T/N expression) value >1 as significantly higher expression, which <−1 as lower expression, and between −1 and 1 as no significant change. T tumorous tissue, N nontumorous tissue. The error bars (**f**) represent standard deviation (s.d) (*n* = 35). ***p < 0.001, paired *t*-test

| Table 1 The relationship of *circTP63* and clinical characteristics in 35 LUSC patients | | | |
|---|---|---|---|
| **Variable** | | *circTP63* | *p*-value |
| | | **High (*n* = 18)** | **Low (*n* = 17)** | |
| Sex | | | | 0.615 |
| | Male | 16 | 15 | |
| | Female | 2 | 2 | |
| Age | | | | 0.065 |
| | ≤60 | 11 | 5 | |
| | >60 | 7 | 12 | |
| Smoke | | | | 0.724 |
| | No | 11 | 10 | |
| | Yes | 7 | 7 | |
| Tumor differentiation | | | | 0.676 |
| | Poorly | 7 | 6 | |
| | Moderately | 4 | 6 | |
| | Well | 7 | 5 | |
| TNM stage | | | | 0.041* |
| | I | 3 | 9 | |
| | II | 5 | 2 | |
| | III | 10 | 6 | |
| Tumor size (cm) | | | | 0.032* |
| | ≤3 | 1 | 5 | |
| | >3 | 17 | 12 | |
| Metastasis | | | | 0.122 |
| | No | 4 | 8 | |
| | Yes | 14 | 9 | |

*LUSC* lung squamous cell carcinoma, *TNM* UICC TNM classification (6th edition)
*p < 0.05, which was considered as a significant difference

was upregulated in LUSC cell lines as compared to human normal lung cell lines (Supplementary Fig. 3a). On the basis of this result, SW900 and H1703 cells were selected for the following *circTP63* loss-of-function assay, whereas H226 and H2170 cells were selected for gain-of-function assay. These results further confirm the characteristics of *circTP63* as a circRNA and imply that its function may be benefited form the biological stability.

**circTP63 promotes cell proliferation and tumor growth**. To study the role of *circTP63* in LUSC progression, we performed short interfering RNAs (si-*circTP63* and si-*circTP63*#2), which specifically target the back-splicing region of *circTP63*. For ectopic overexpression of *circTP63*, exon 10 and 11 of *TP63* were cloned into the lentiviral vector (Supplementary Fig. 3b). We found that *circTP63* siRNAs could successfully knockdown *circTP63* expression but had no effect on *TP63* mRNA expression in SW900 and H1703 cells (Fig. 3a and Supplementary Fig. 3c). Similarly, *circTP63* was successfully overexpressed in H226 and H2170 cells, while *TP63* mRNA expression had no obvious change (Fig. 3b). These data indicated that the expression of *TP63* was unaffected by *circTP63*. Cell proliferation assay revealed that silencing of *circTP63* significantly suppressed cell growth (Fig. 3c

and Supplementary Fig. 3d). Conversely, stably overexpressing *circTP63* remarkably promoted cell viability (Fig. 3d). In order to prove that *circTP63* was responsible for the phenotypes, we mutated the si-*circTP63* targeted back splice junction (Supplementary Fig. 3b). The *circTP63*-si-mut plasmid was co-transfected with si-*circTP63* into H1703 cells to determine whether it could affect the cell proliferation. Result showed that *circTP63*-si-mut could rescue the proliferation phenotype and promoted the cell growth (Supplementary Fig. 3e). Moreover, due to the proliferation-promoting effect of *TP63* (Supplementary Fig. 3f), we performed another rescue experiment to show that the function of *circTP63* was independent of *TP63* (Supplementary Fig. 3g). Cell cycle analysis illustrated that silencing of *circTP63* decreased the number of cells in G2/M phase, but increased the number of cells in G1 phase as compared with the controls (Fig. 3e and Supplementary Fig. 4a). Ectopic expression of *circTP63* led to the progression of cells from the G1/S to G2/M phase (Fig. 3f and Supplementary Fig. 4b), which suggested an increase in cell cycle progression. However, ectopic expression of *circTP63* had no significant effect on migration and invasion of H2170 and H226 cells (Supplementary Fig. 4c).

To identify the effect of *circTP63* on tumor growth in vivo, we established a nude mice xenograft model by implanting H2170 cells with vector or *circTP63*. The tumor volumes were monitored from the 14 days after H2170 cell injection. We found that overexpression of *circTP63* drastically increased tumor growth of H2170 cells. The tumor volumes and weights were significantly accelerated by *circTP63* (Fig. 3g). In addition, the impact of *circTP63* knockdown upon tumor growth in vivo was also investigated. A xenograft tumor model of H1703 cells was developed, then treated with intratumoral injection of cholesterol-conjugated si-*circTP63* or si-NC. As shown in Fig. 3h, treatment with si-*circTP63* significantly inhibited growth of H1703 in vivo. After harvesting the subcutaneous tumour tissues, immunohistochemistry was performed. Results revealed that xenograft tumors derived from H1703 cells with *circTP63* knockdown had lower expression of Ki67 and PCNA than the si-NC group (Supplementary Fig. 4d), and the expression of *circTP63* in xenograft tumors were confirmed (Supplementary Fig. 4e). Taken together, these findings suggest that *circTP63* may play an oncogenic role in LUSC in vitro and in vivo.

**circTP63 facilitates cell proliferation by targeting FOXM1**. We analyzed co-expression patterns of *circTP63* and mRNAs according to the results of the SBC Human ceRNA Array analysis. In the co-expression network, 25 putative mRNAs were selected based on the rank of Pearson correlation coefficient values (≥0.95) (Fig. 4a). Of these, *FOXM1*, *KIF18B*, and *BRCA1* were the top three predicted co-expressed mRNAs (Supplementary Table 2). Microarray data showed 25 putative mRNAs were significantly upregulated in LUSC tissues and *FOXM1* was upregulated by more than eight-fold in LUSC (Fig. 4b and Supplementary Table 3). We further confirmed the upregulated level of *FOXM1* in 35 paired LUSC samples (Fig. 4c). Meanwhile, we

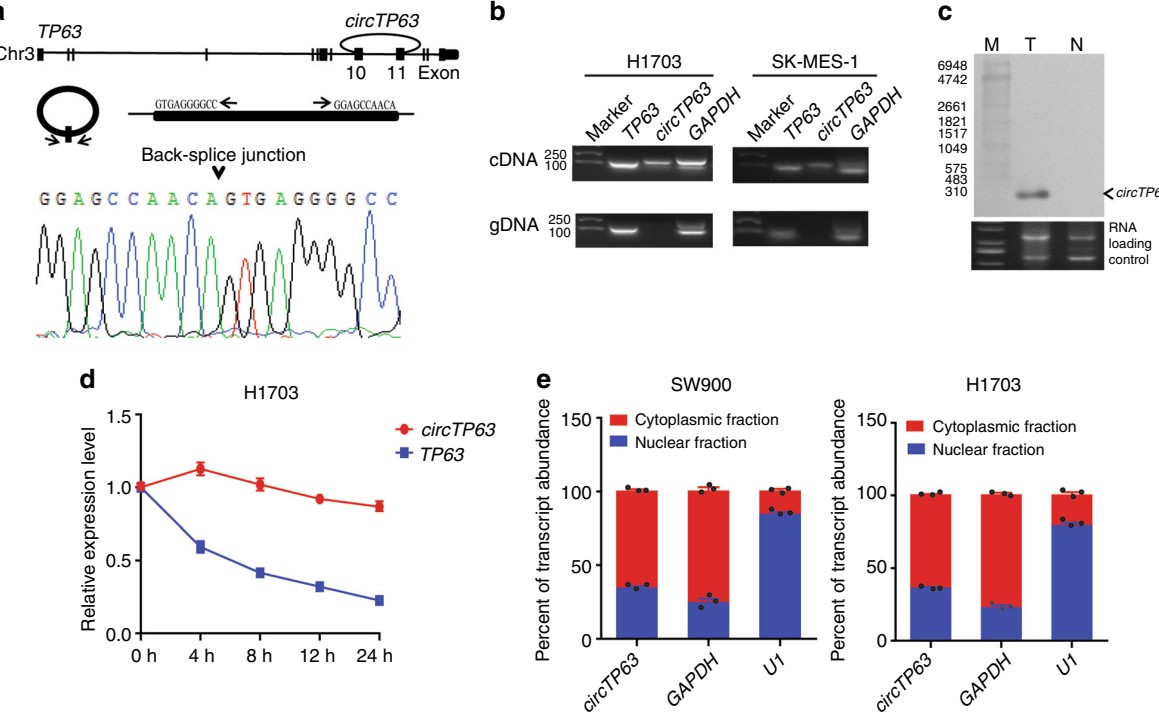

**Fig. 2** Characterization of *circTP63* in LUSC cells. **a** Genomic loci of *circTP63* gene. *circTP63* is produced at the *TP63* gene (NM_003722.4) locus containing exons 10–11. The back-splice junction of *circTP63* was identified by Sanger sequencing. **b** PCR analysis for *circTP63* and its linear isoform *TP63* in cDNA and genomic DNA (gDNA). **c** Northern blot analysis showed the size and abundance of *circTP63* in one paired sample of LUSC tumorous tissue and corresponding adjacent nontumorous tissues. M: marker. **d** qRT-PCR for the abundance of *circTP63* and *TP63* in H1703 cells treated with Actinomycin D at the indicated time point. **e** Levels of *circTP63* in the nuclear and cytoplasmic fractions of SW900 and H1703 cells. The error bars (**d**, **e**) represent s.d. ($n = 3$)

detected the correlation of *circTP63* and these three genes in 35 LUSC tissues. Result showed that *circTP63* expression was most positively correlated with *FOXM1* compared to *KIF18B* or *BRCA1* (Fig. 4d and Supplementary Fig. 5a). Although *KIF18B* may be also regulated by *circTP63* (Supplementary Fig. 5b), the effect was still weaker than *FOXM1*. For *BRCA1*, although there was a significant change in H2170 cells with *circTP63* overexpression (Supplementary Fig. 5b), no significant correlation between expression of *circTP63* and *BRCA1* in LUSC tissues was observed (Supplementary Fig. 5a). Analysis of the protein level of FOXM1 in eight paired LUSC samples and the correlation with the transcriptional level of *circTP63* showed that FOXM1 was significantly upregulated in LUSC tissues and positively correlated with *circTP63* (Fig. 4e). qRT-PCR and western blots revealed that *circTP63* knockdown significantly reduced the levels of *FOXM1* mRNA and protein, and the opposite results were observed when *circTP63* was overexpressed (Fig. 4f). Therefore, FOXM1 was considered as a major candidate target of *circTP63*.

To explore the function of *circTP63* on FOXM1, cell proliferation after knockdown of *FOXM1* was examined. When mRNA and protein levels of *FOXM1* were reduced at least 50% by *FOXM1* siRNAs (Supplementary Fig. 5c), the growth of H2170 cells was significantly inhibited (Supplementary Fig. 5d), and cell cycle showed that less number of cells in G2/M phase (Supplementary Fig. 5e). In addition, knockdown of *FOXM1* could abrogate the effects of *circTP63* on promoting cell proliferation (Fig. 4g), while overexpression of *FOXM1* could significantly rescue the proliferative inhibition of H1703 cells with *circTP63* knockdown (Fig. 4h) and promoted cell cycle progression from G1/S to G2/M phase (Supplementary Fig. 5f). These results indicate that *circTP63* may be capable of modulating proliferation of LUSC cells by targeting FOXM1.

**circTP63 relieves repression of miR-873-3p on FOXM1**. It has been shown that circRNAs can act as miRNAs sponge to regulate downstream targets[16,19–21]. Given that *circTP63* mainly located in the cytoplasm (Fig. 2d), we explored whether *circTP63* might also function as the ceRNA mechanism. Based on the theory of ceRNA[33], circRNAs can share the same miRNAs with mRNA. Therefore, we constructed a *circTP63-miRNAs-FOXM1* network by miRanda prediction. This network included 22 candidate miRNAs (Supplementary Fig. 6a), containing common binding sites for the *circTP63* and *FOXM1* (Fig. 5a and Supplementary Table 4). To validate binding capability of the miRNAs to *circTP63*, we constructed the *circTP63* luciferase reporter system. Each predicted miRNA mimics was co-transfected with *circTP63* luciferase reporter into HEK-293T cell. We observed that multiple miRNAs were able to reduce luciferase activity and *miR-873–3p* reduced most by at least 80% (Fig. 5b). We next preformed AGO2 immunoprecipitation to determine whether *circTP63* served as a platform for AGO2 and *miR-873-3p*. As shown in Fig. 5c, *circTP63* was specifically enriched in *miR-873-3p* transfected cells. To confirm *circTP63* and *FOXM1* could be regulated by *miR-873-3p*, we constructed luciferase reporters containing wild type and mutated putative binding sites of *circTP63* or *FOXM1* transcripts (Supplementary Fig. 6b), respectively. Luciferase reporter assays showed that the luciferase activities of *circTP63* or *FOXM1* wild type reporter were significantly reduced when transfected with *miR-873-3p* mimics compared with control reporter or mutated luciferase reporter (Fig. 5d). In addition, we found that *circTP63* overexpression or knockdown could further increase or reduce the luciferase activity of *FOXM1* wild type reporter (Fig. 5e). Analyses of the mRNA and protein levels of FOXM1 showed that *miR-873-3p* inhibitors significantly increased *FOXM1* mRNA and protein levels in

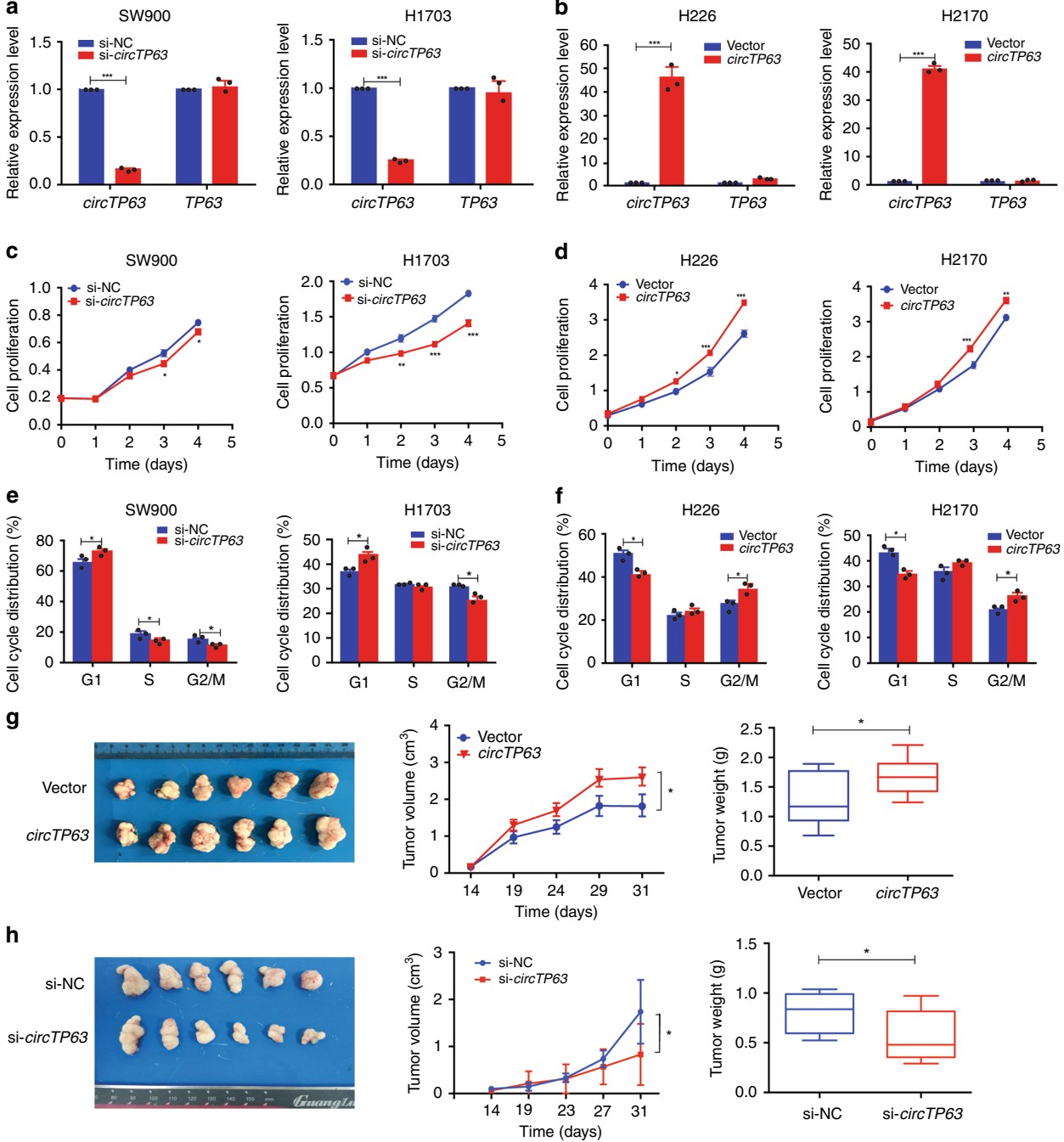

**Fig. 3** circTP63 promotes cell proliferation and tumor growth both in vitro and in vivo. **a** Expression levels of circTP63 and TP63 in SW900 and H1703 cells treated with circTP63 siRNA. **b** Expression levels of circTP63 and TP63 in H226 and H2170 cells after transduction with circTP63 lentivirus. **c** and **d** Cell proliferation analysis of LUSC cells with silencing or stably overexpressing circTP63. **e** and **f** Cell cycle analysis of LUSC cells with silencing or stably overexpressing circTP63. **g** The volume and weight of subcutaneous xenograft tumors of H2170 cells isolated from nude mice. **h** The volume and weight of subcutaneous xenograft tumors of H1703 cells isolated from nude mice; center line: median of data; Bounds of box: the second quartile to the third quartile; Whisker: minimum value to maximum value. The error bars **a–h** represent s.d. (in **a–f**, $n = 3$; in **g** and **h**, $n = 6$). $*p < 0.05$; $**p < 0.01$; $***p < 0.001$, two-tailed $t$-test. Source data are provided as a Source Data file

SW900 and H1703 cells (Supplementary Fig. 6c). These results suggest that circTP63 can bind to miR-873-3p, and FOXM1 can be regulated by miR-873-3p and circTP63.

In order to further confirm that circTP63 can serve as a ceRNA to regulate FOXM1 expression, we measured the absolute expression levels of circTP63 and miR-873–3p in LUSC cell lines by copy numbers. Results showed that the absolute expression of circTP63 is much higher than miR-873-3p in most of tested LUSC

cell lines (Supplementary Fig. 6d), suggesting that circTP63 is rational to sponge miR-873-3p. Then we used biotin-coupled miR-873-3p mimics for pull down assay to detect competitive binding activities of circTP63 and FOXM1 to miR-873-3p in H2170 cells. We noted that a nearly three-fold enrichment of circTP63 and a nearly six-fold enrichment of FOXM1 in the miR-873-3p captured fraction compared with the negative control (Fig. 5f, upper panel), and overexpression of circTP63 in H2170

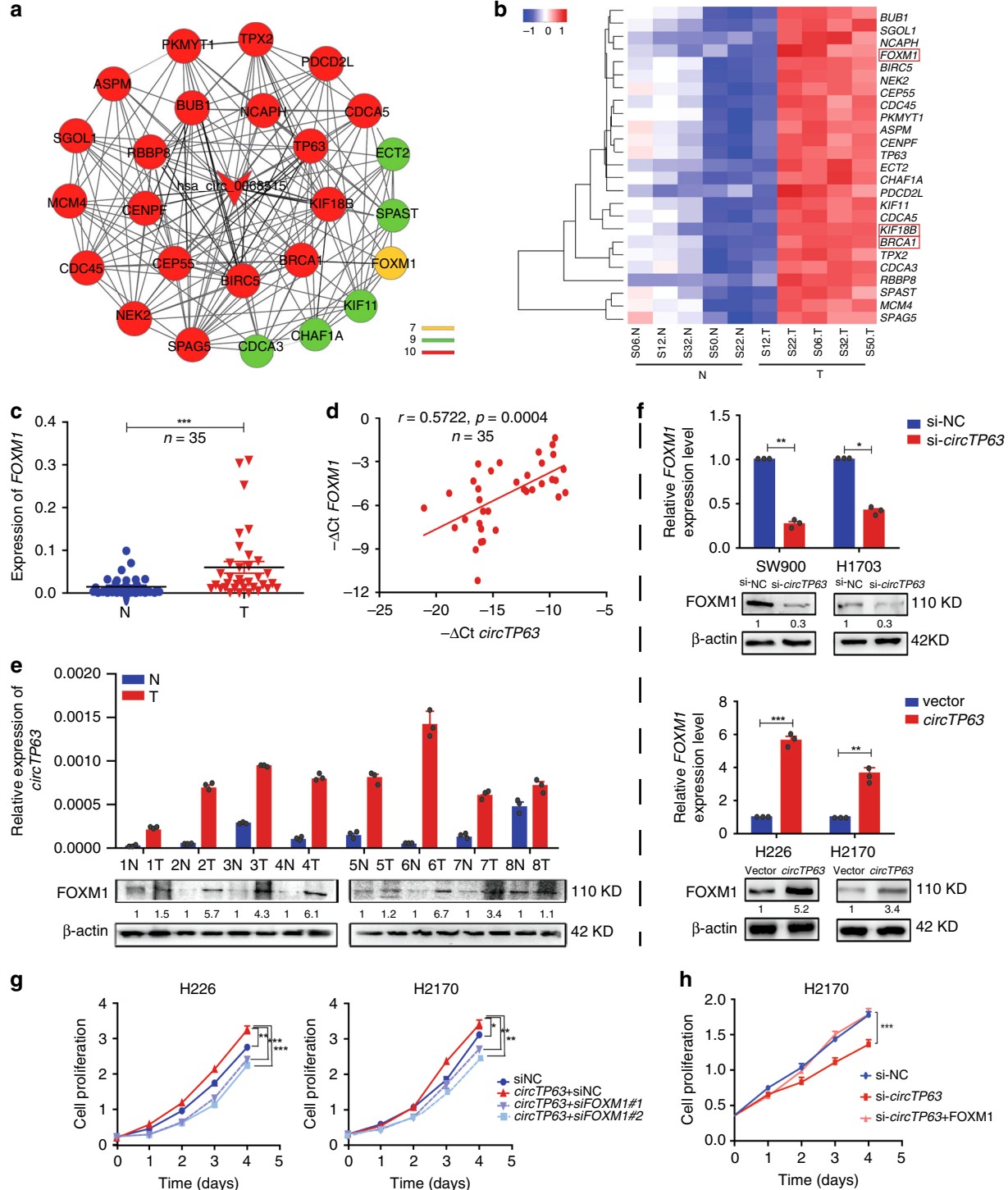

**Fig. 4** circTP63 contributes to cell proliferation through targeting FOXM1. **a** Co-expression network of circTP63 with associated 25 mRNAs. A round node represents a protein-coding gene and the arrow node represents circTP63 (hsa_circ_0068515). Lines between two nodes indicate interactions between two genes. Color represents the number of lines. **b** A heatmap shows mRNA levels of these 25 co-expression genes in the five paired LUSC samples of SBC Human ceRNA Array analysis. **c** Expression analysis for FOXM1 in additional 35 paired LUSC samples. **d** Correlation analysis revealed positive correlation between the levels of circTP63 and FOXM1 mRNA in the tumorous tissues of the 35 LUSC patients. ΔCt values were normalized according to β-actin. **e** The levels of circTP63 expression and FOXM1 protein in eight paired LUSC samples. **f** The mRNA and protein levels of FOXM1 in the LUSC cells with knockdown or overexpression of circTP63. **g** Cell proliferation assay for H226 and H2170 cells with circTP63 overexpression and FOXM1 knockdown. **h** Cell proliferation assay for H1703 cells with circTP63 knockdown and FOXM1 overexpression. The error bars **c**, **e**–**h** represent s.d. (in **c**, n = 35; in **e**–**h**, n = 3). *p < 0.05; **p < 0.01; ***p < 0.001, two-tailed t-test. Source data are provided as a Source Data file

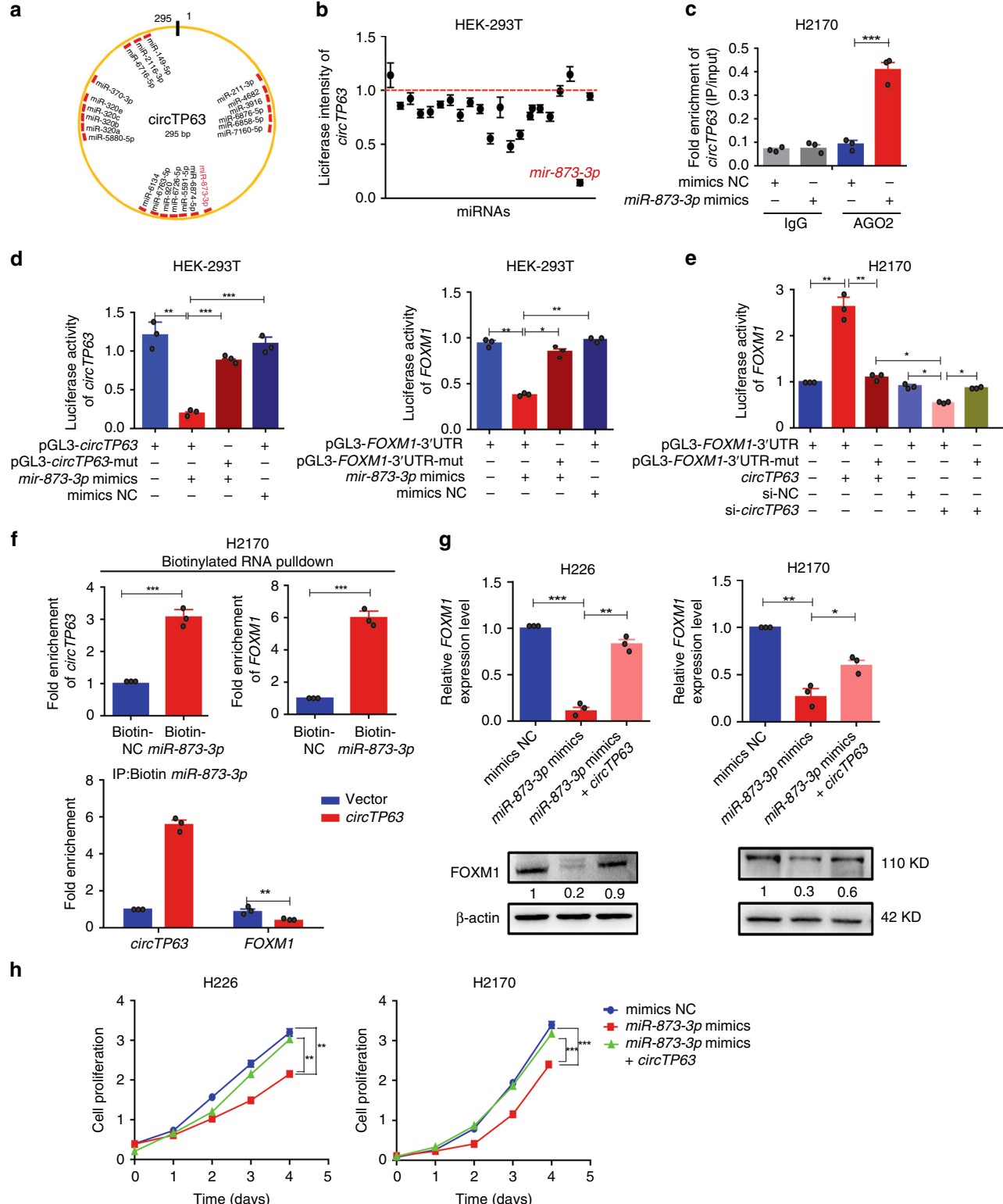

cells led to the decreased enrichment of *FOXM1* on *miR-873-3p* (Fig. 5f, lower panel). Additionally, we also evaluated *FOXM1* in cells with overexpressing *miR-873-3p*. Result showed that *miR-873-3p* mimics significantly decreased *FOXM1*, but it was rescued by *circTP63* overexpression (Fig. 5g). We investigated the effect of *miR-873-3p* on cell proliferation. Overexpression of *miR-873-3p* inhibited the cell proliferation and overexpression of *circTP63* could rescue *miR-873-3p* mimics-mediated suppression for proliferation and cell cycle (Fig. 5h and Supplementary Fig. 6e).

To futher confirm the cellular phenotype was caused by binding of *circTP63* with *miR-873-3p*, we mutated *miR-873-3p*-binding site on *circTP63* and transfected the *circTP63*-miR-mut into H226 and H2170 cells. Results showed there was no significant difference in proliferation between cells transfected with *circTP63*-miR-mut or control vector (Supplementary Fig. 6f). These results reveal that *circTP63* servers as a sponge for *miR-873-3p* to regulate FOXM1 and promoted cell proliferation via the ceRNA mechanism in LUSC cells.

**Fig. 5** *circTP63* facilitates cell proliferation by relieving repression of *miR-873-3p* for FOXM1 expression. **a** A schematic model shows the putative binding sites of 22 predicted miRNAs on *circTP63*. **b** Luciferase activity of *circTP63* in HEK293T cells transfected with miRNA mimics which are putative binding to the *circTP63* sequence. Luciferase activity was normalized by Renila luciferase activity. **c** RIP was performed using AGO2 antibody in H2170 cells transfected with *miR-873-3p* mimics or mimics NC, then the enrichment of *circTP63* was detected. **d** Luciferase reporter activity of *circTP63* (left) and *FOXM1*-3′UTR (right) in HEK-293T cells co-transfected with *miR-873-3p* mimics or mimics NC. **e** Luciferase reporter activity of *FOXM1*–3′UTR in H2170 cells with *circTP63* knockdown or overexpression. **f** Upper panel: *circTP63* and *FOXM1* were pulled down and enriched with 3′-end biotinylated *miR-873-3p*. Lower panel: binding activities of *circTP63* and *FOXM1* to 3′-end biotinylated *miR-873-3p* in H2170 cells with *circTP63* overexpression. **g** *FOXM1* expression in H226 and H2170 cells transfected with *miR-873-3p* mimics alone or co-transfected with *circTP63*. **h** Cell proliferation analysis for H226 and H2170 cells transfected with *miR-873-3p* mimics alone or co-transfected with *circTP63*. The error bars **b**–**h** represent s.d. ($n = 3$). $*p < 0.05$; $**p < 0.01$; $***p < 0.001$, two-tailed *t*-test. Source data are provided as a Source Data file

**CENPA and CENPB are regulated by *circTP63* through FOXM1**. FOXM1 is previously identified as an important cell cycle regulator controlling transition from G1 to S phase and cellular progression into the M phase[34]. According to previous reports[35,36], we selected eight cell cycle-related candidates (*AURKA, AURKB, CDC25B, CENPA, CENPB, CENPF, PLK1,* and *CCNB1*), which are directly targeted by *FOXM1*. mRNA levels of the eight candidates were detected after *circTP63* overexpression. Results showed that *CENPA, CENPB,* and *CCNB1* were regulated by *circTP63*, while another 5-cell cycle-related genes had no significant change (Fig. 6a). Then we designed rescue experiments. *FOXM1* siRNAs significantly attenuated the effects of *circTP63* on *CENPA* and *CENPB* (Fig. 6b). We knocked down *CENPA* and *CENPB* in H226 and H2170 cells with *circTP63* overexpression. Cell proliferation analysis showed that knockdown of *CENPA* or *CENPB* alone could diminish the effect of *circTP63* overexpression on proliferation (Supplementary Fig. 7a, b). And when *CENPA* and *CENPB* were knocked down together, more remarkable suppression effect on cell proliferation was observed (Fig. 6c). These results proved that knockdown of *CENPA* and *CENPB* could mimic the phenotypes of si*FOXM1* in *circTP63*-expressing cells.

We further analyzed the expression of *FOXM1, CENPA,* and *CENPB* in xenograft tumors by qRT-PCR. Results showed lower expression of *FOXM1, CENPA,* and *CENPB* in si-*circTP63* group than in si-NC group (Supplementary Fig. 7c). In addition, we detected the expression of *CENPA* and *CENPB* in the 35 paired samples of LUSC by qRT-PCR and analyzed the correlation between *circTP63* with *CENPA* or *CENPB*. Results showed that the expression of *CENPA* and *CENPB*, particularly *CENPA*, was increased in LUSC tissues as compared to their corresponding adjacent nontumorous tissues (Supplementary Fig. 7d), and positively related to the expression of *circTP63* (Supplementary Fig. 7e). These data indicate that *circTP63* promotes the cell proliferation through the FOXM1-CENPA/B pathway.

## Discussion

circRNA expression profiling is a prerequisite for the identification of novel tumor suppressors and oncogenic circRNAs, as well as in elucidating their mechanisms and functions[28]. In this study, a co-expression profiling of circRNA and mRNA in LUSC was elucidated by global different expression microarray analysis. Based on the co-expression profiling analyses of circRNAs and mRNAs, we identified *circTP63* as a significantly upregulated circRNA in LUSC tissues. Gain-of-function and loss-of-function experiments demonstrated that *circTP63* was associated with cell cycle and proliferation. *circTP63* exerted its function as a ceRNA that competitively bound to *miR-873-3p*, then abolished the endogenous suppressive effect of *miR-873-3p* on the target gene *FOXM1*. Elevated *FOXM1* could promote the expression of *CENPA* and *CENPB*, then drive cell cycle progression and cell proliferation (Fig. 6d), revealing that *circTP63* promotes LUSC cell growth via the ceRNA mechanism.

Genetic alterations in signaling networks frequently occur in cancer to sustain proliferation and maintain viability[37]. Cell cycle alterations are common in almost all cancer types. In our study, we observed that differentially expressed mRNAs were mainly enriched in cell cycle pathway (Fig. 1c). Noncoding RNAs are often reported to be associated with the regulation of cell survival[38]. We did co-expression network analysis with dysregulated circRNAs and key genes in cell cycle to filter candidate circRNAs (Supplementary Fig. 2a). Among them, we characterized that *circTP63* was expressed at low level in nontumorous tissues but at high level in LUSC tissues (Fig. 1f) and could promote cell proliferation and cell cycle progression in vitro and in vivo (Fig. 3), indicating that *circTP63* is a potential cancer-related gene in LUSC.

FOXM1, human Forkhead Box M1, is identified to be highly expressed in various cancers, such as lung cancer[39], glioblastomas[40], basal cell carcinomas[41], infiltrating ductal breast carcinomas[42], and intrahepatic cholangiocarcinomas[43]. In this study, we showed that *FOXM1* was upregulated in LUSC tissues and knockdown of *FOXM1* inhibited the cell proliferation. Notably, *circTP63* could upregulate *FOXM1* expression and the effect of *circTP63* on promoting cell proliferation could be blocked by silencing *FOXM1* (Fig. 4). Furthermore, elevated expression of *FOXM1* in NSCLC is significantly associated with higher TNM stage, advanced tumor size, and poor prognosis[44], which supports that high level of *circTP63* is associated with larger tumor size and severer TNM stage (Table 1).

The ceRNA hypothesis has proposed that RNA transcripts, such as mRNAs, lnRNAs, and circRNAs share the miRNA response elements, competing for binding the miRNAs, and then regulating the expression of each other, constructing a complex post-transcriptional regulatory network[33]. In our study, we performed bioinformatic analyses to select miRNAs, which shared common binding sites with *circTP63* and *FOXM1*. Simultaneously, we designed *circTP63* luciferase reporter screening for these miRNAs. We found that *miR-873-3p* reduced the luciferase activity of *circTP63* luciferase reporter most by at least 80% (Fig. 5b). Considering the strongest binding strength with *circTP63*, *miR-873-3p* was verified as the binding target of *circTP63*. On the other hand, our current study unveiled that *FOXM1* was the direct target of *miR-873-3p* (Fig. 5d, g and Supplementary Fig. 6c). All of the above results suggest *miR-873-3p* can bind with *circTP63* and *FOXM1*, respectively. Subsequent biotinylated miRNA pull-down assays showed the competitive binding activities of *circTP63* and *FOXM1* to *miR-873-3p* (Fig.5f) and further rescue experiment showed that *circTP63* significantly attenuated the effects of *miR-873-3p* on *FOXM1* (Fig. 5g–h), suggesting that *circTP63* may function as a ceRNA to regulate FOXM1 in LUSC.

*miR-837* has been reported to play an important role in post transcriptional regulation of several diseases. For example, *miR-837* suppressed cell proliferation and tumorigenesis in glioblastoma cancer[45] and induced necrosis in cardiovascular

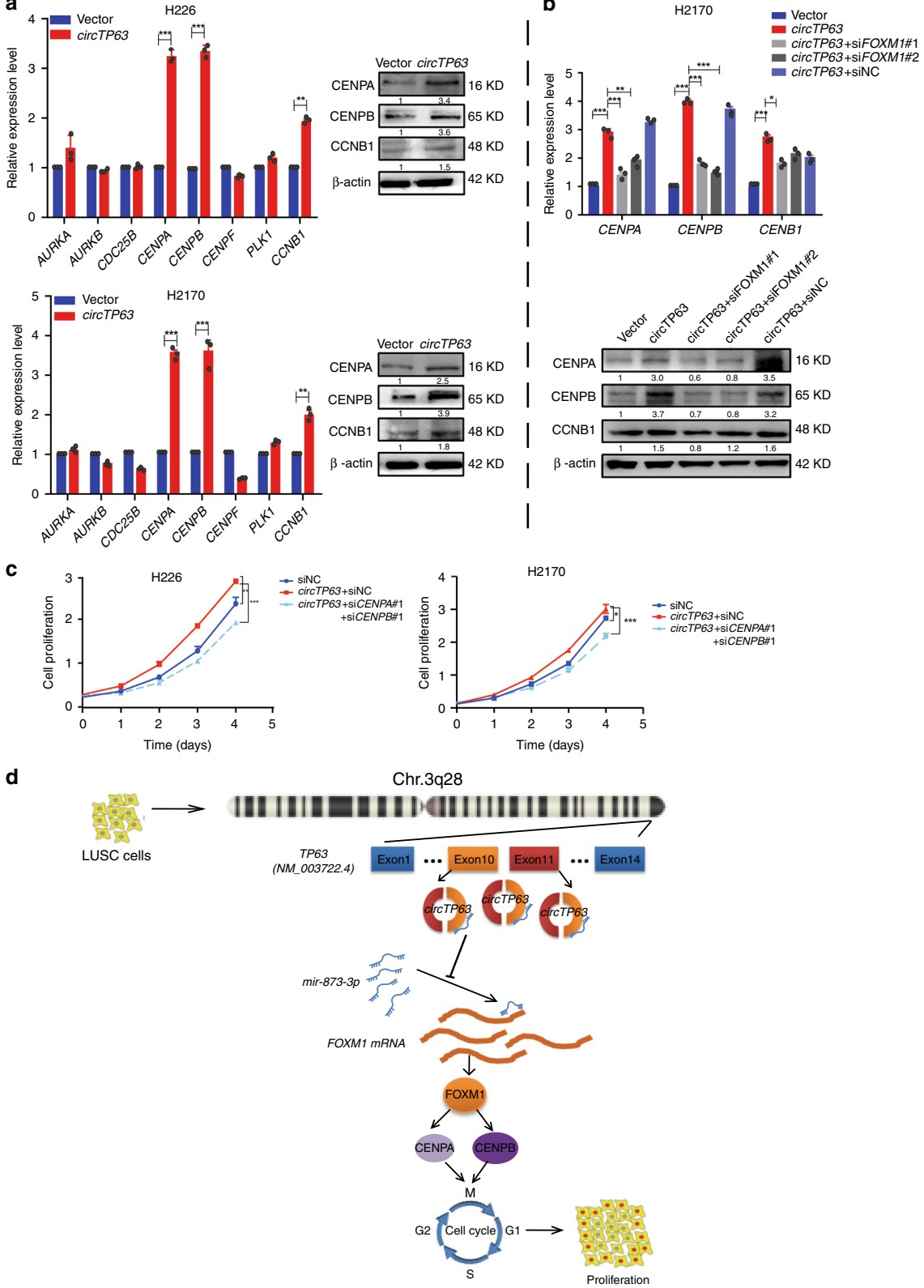

**Fig. 6** CENPA and CENPB are regulated by *cicrTP63* through FOXM1. **a** The mRNA and protein levels of cell cycle-related genes in H226 and H2170 cells with *circTP63* overexpression. **b** Expression changes of *CENPA*, *CENPB*, and *CCNB1* after knockdown of *FOXM1* in H2170 cells with *circTP63* overexpression. **c** Cell proliferation assay for H226 and H2170 cells with *circTP63* overexpression and joint knockdown of *CENPA* and *CENPB*. **d** Hypothesis diagram illustrates function and mechanism of *circTP63* in LUSC progress. The error bars **a–c** represent s.d. ($n = 3$). *$p < 0.05$; **$p < 0.01$; ***$p < 0.001$, two-tailed *t*-test. Source data are provided as a Source Data file

disease[46]. However, the roles of *miR-873-3p* in LUSC remain unclear. In this study, we found that *miR-873-3p* could inhibit cell proliferation in LUSC (Fig. 5h). The abundance of *miR-873-3p* was lower of *circTP63* in LUSC cell lines (Supplementary Fig. 6d) and *miR-873-3p* was sponged by *circTP63* (Fig. 5b), resulting in less *miR-873-3p* to target *FOXM1* and increase cell proliferation. Identification of *circTP63-miR-873–3p-FOXM1* axis expands the understanding of the underlying mechanism of LUSC progression.

Finally, we explored downstream targets of *FOXM1* essential for *circTP63*-mediated oncogenic function. Overexpression of *circTP63* increased the expression of *CENPA* and *CENPB* through upregulating *FOXM1* (Fig. 6a, b). CENPA, histone H3 variant, is the key determinant of centromere identity[47]. CENPB, heterochromatin protein, directly binds amino-terminal tail of CENPA to enhance the fidelity of human centromere function[48]. They are induced in G1 phase and actived in M phase[49]. It has been reported that *CENPA* was highly expressed in lung cancer tissue and associated with poorer overall survival[50]. However, why *CENPA* was upregulated in lung cancer has not been explored. In this study, our results partially uncover that *CENPA* can be regulated by *circTP63* in LUSC.

In conclusion, our study reveals that *circTP63* competitively binds *miR-873-3p* to abolish the suppressive effect of *miR-873-3p* on FOXM1, then promotes cell proliferation. Our findings provide an insight into understanding the development and progression of LUSC, and a potential therapeutic approach for LUSC.

## Methods

**Tissues and cell lines**. In total, 40 paired samples of tumorous tissues (T) and adjacent nontumorous tissues (N) were obtained from surgical resections of LUSC patients without preoperative treatment at Shanghai Chest Hospital (Shanghai, China). The samples were identified by two pathologists independently. Among them, five paired samples of tumorous tissues and adjacent nontumorous tissues were used for microarray analysis, and 35 paired samples for qRT-PCR verification. These samples were all stored at −80 °C until use. The detailed clinicopathological features are described in Table 1 and Supplementary Tables 5 and 6. All tissue specimens were collected from July 2013 to September 2014. The human materials were obtained with the consent of patients and approved by Ethics Committees of the Shanghai Chest Hospital and Shanghai Cancer Institute.

All of human LUSC cells (NCI-H2170, NCI-H1703, NCI-H226, NCI-H520, SW900, SK-MES-1, BEAS-2B, and HFL-1) were purchased from the American Type Culture Collection (ATCC) and were tested negative for mycoplasma contamination. NCI-H2170, NCI-H1703, NCI-H226, and NCI-H520 cells were cultured in RPMI-1640 medium with 10% FBS; SK-MES-1 cell was cultured in MEM medium with 10% FBS; HFL-1 cell was cultured in F-12K medium with 10% FBS and BEAS-2B was cultured in BEGM medium. They were all cultured at 37 °C with 5% $CO_2$. SW900 cell was cultured in Leibovitz's L-15 medium with 10% FBS in a free gas exchange with atmospheric air.

**Microarray analysis**. Total RNAs were isolated from the paired tissue samples of five LUSC patients by TRIzol reagent (Invitrogen) and purified by RNeasy Mini Kit (Qiagen). RNA samples were then used to generate fluorescence-labeled cRNA targets for the SBC human ceRNA array V1.0 (4 × 180K, designed by Shanghai Biotechnology corporation, and made by Agilent technologies), which contains 88,371 circRNA probes, 77,103 lncRNA probes, and 18,853 mRNA probes. The labeled cRNA targets were then hybridized in the slides. After hybridization, slides were scanned on the Agilent Microarray Scanner. Data were extracted with Feature Extraction software 10.7 (Agilent technologies). Raw data were normalized by Quantile algorithm, limma package the R program. Significant differential expressed transcripts were screened by fold change ≥2 or ≤−2 and *p*-value ≤ 0.05.

**Co-expression and ceRNA analysis for *circTP63***. The circRNAs/mRNAs co-expression analysis was based on calculating the Pearson correlation coefficient (PCC) between the expression levels of mRNA and circRNA in the SBC Human ceRNA Array analysis. The value of parameter PCC ≥ 0.95 and *p*-value < 0.01 was recommended for further analysis.

The ceRNA analysis (*circTP63/miRNAs/FOXM1* interaction) was predicted by miRanda and identified binding sites with relatively high scores (≥140). The networks were visualized by Cytoscape.

**RNA and gDNA extraction**. Total RNAs were extracted from cells using Trizol reagent (Invitrogen) according to the manufacturer's instruction. The nuclear

and cytoplasmic fractions were extracted using PARIS Kit (Ambion, Life Technologies). gDNA was extracted using Genomic DNA Isolation Kit (Sangon Biotech, Shanghai, China).

**RT-PCR and qRT-PCR**. RNA was reverse transcribed using HiScript II Q RT SuperMixfor qPCR (+gDNA wiper) (Vazyme, Nanjing, China). The AmpliTaq DNA Polymerase (Life Technologies) was used for PCR. The cDNA and gDNA PCR products were observed using 2% agarose gel electrophoresis. AceQ qPCR SYBR Green Master Mix (Vazyme, Nanjing, China) was used for qRT-PCR. The circRNA and mRNA levels were normalized by *β-actin*. The miRNA level was normalized by small nuclear *U6*. The relative expression levels were determined by the $2^{-\Delta Ct}$ or $2^{-\Delta\Delta Ct}$ method. To determine the absolute quantity of RNA, purified RT-PCR products were used to generate the standard curve. Briefly, *circTP63* and *miR-873–3p* form cDNAs were amplified, purified and measured. Then they were serially diluted to be as templates for qRT-PCR. The standard curves were drawn according to the Ct values at different concentrations. According to the standard curves, copy numbers of *circTP63* and *miR-873–3p* in six LUSC cell lines were calculated. Primers are listed in Supplementary Table 7.

**RNase R treatment**. Two micrograms of total RNA was incubated for 30 min at 37 °C with or without 5 U/μg RNase R (Epicentre Technologies), and subsequently purified by RNeasy MinElute Cleaning Kit (Qiagen), then analyzed by RT-PCR.

**Actinomycin D assay**. H1703 cells were exposed to 2 μg/ml actinomycin D (Sigma) at indicated time point. Then the cells were harvested, and total RNA was extracted. The stability of *circTP63* and *TP63* mRNA was analyzed using qRT-PCR.

**Vector construction and cell transfection**. A sketch map was drawn to show how to make recombinants for *circTP63* overexpression or *circTP63*-si-mut (back-splice junction mutant) overexpression (Supplementary Fig. 3b). Briefly, for *circTP63* overexpression version, a sequence which orderly contains EcoRI site, cyclization-mediated sequence-F, splice acceptor AG, linear *TP63*, splice donor GT, cyclization-mediated sequence-R, and BamHI site was amplified by PrimerSTAR Max DNA Polymerase Mix (Takara). Then the PCR product was inserted into PLCDH-ciR which was reconstructed by inserting front circular frame and back circular frame to promote RNA circularization. For the mutant version of *circTP63*, the same method was performed except using different primers. In mutant primer-F, GCCAACA (the sequence in the 5′ end of *TP63* mRNA that is targeted by si-*circTP63*) was replaced with ACAACCG. Similarly, GTGAGGGGCCGT (the sequence in the 3′end of *TP63* mRNA that is targeted by si-*circTP63*) was replaced with TGCCGGGGAGTG in mutant primer-R. As a result, when the recombinant plasmids are transfected into cells, the spliceosome recognizes the AG splice acceptor and GT splice donor. The linear *TP63* or mutant linear *TP63* is back spliced, and generated as *circTP63* or *circTP63*-si-mut. For luciferase reporter vector, the sequence of *circTP63* and *FOXM1* 3′UTR was cloned into the downstream of pGL3-promoter. Mutations of miRNA-binding sites in *circTP63* and *FOXM1* 3′UTR sequence were generated using Mutagenesis Kit (Vazyme, Nanjing, China). siRNAs of *circTP63* and *FOXM1*, *miR-873-3p* mimics and inhibitors, and corresponding negative control (NC) were synthesized by GenePharma (Shanghai, China). Cells were transfected using Lipofectamine 2000 (Invitrogen) and harvested for experiment after 48 h. Primers and oligonucleotide sequence are listed in Supplementary Table 7.

**CCK-8 assay and cell cycle analysis**. For the cell proliferation assay, $1 \times 10^3$ cells were seeded in 100 μl of complete culture media in 96-well plates for various time periods. Cell Counting Kit-8 assay (Dojindo Laboratories, Kumamoto, Japan) was performed to measure cell viability according to manufacturer's instructions. For cell cycle analysis, $1 \times 10^5$ cells were labeled with PI/RNase Staining Buffer (BD Bioscience) according to the manufacturer's instructions. The DNA content was determined using flow cytometry (Beckman FC500, Los Angeles, CA, USA) and analyzed by Modfit software.

**Xenograft model**. For H2170 xenograft model: 6–8-week-old male BALB/c nude mice were housed under standard conditions and cared for according to protocols. $2 \times 10^6$ H2170 cells with *circTP63* overexpressed vector or control vector were suspended in 200 μl serum-free RPMI-1640 and subcutaneously injected into the right flank of each mouse. The volumes of tumors were measured from 14 days after injecting. After 31 days the mice were sacrificed.

For xenograft model of H1703 cells: $2 \times 10^6$ H1703 cells were subcutaneously injected into a single flank of each mouse (12 mice in total). Two weeks later, mice with palpable tumors (~62 $mm^3$) were randomly divided into two groups (six mice per group), 50 nmol cholesterol-conjugated si-NC or si-*circTP63* was intratumorally injected into the two groups three times per week for two weeks. Tumor growth was examined every 4–5 days. After mice were sacrificed, tumors were weighed and processed for further histological analysis.

Tumor volume was calculated as follows: $V$ (volume) = (length × width$^2$)/2.

All animal experiments were performed under approval by the Shanghai Medical Experimental Animal Care Commission.

**Luciferase activity assays**. HEK-293T cells were seeded in 96-well plates at a density of $5 \times 10^3$ cells per well for 24 h before transfection. The cells were co-transfected with a mixture of 50 ng luciferase reporter vectors, 5 ng Renila luciferase reporter vectors (pRL-TK), and miRNA mimics at the indicated concentration. After 48 h, the luciferase activity was measured with a dual luciferase reporter assay system (Promega). The luciferase values were normalized to the corresponding Renila luciferase values, and then the fold changes were calculated.

**Biotin-coupled miRNA capture**. The 3′end biotinylated *miR-873-3p* mimics or control RNA (Ribio, Guangzhou, China) were transfected into $1 \times 10^6$ H2170 cells at a final concentration of 50 nM for 48 h before harvest. Then 0.7 ml lysis buffer (5 mM $MgCl_2$, 100 mM KCl, 20 mM Tris (pH 7.5), 0.3% NP-40, 50U of RNase OUT (Invitrogen, USA)) and complete protease inhibitor cocktail (Roche Applied Science, IN) were added into the cell pellets, and incubated on ice for 10 min. The biotin-coupled RNA complex was pulled down by incubating the cell lysates with streptavidin-coated magnetic beads (Life Technologies) by centrifugation at $10,000 \times g$ for 10 min. The abundance of *circTP63* in bound fraction was evaluated by qRT-PCR analysis.

**RNA immunoprecipitation (RIP)**. RIP experiments were performed with a Magna RIP RNA-Binding Protein Immunoprecipitation Kit (Millipore, Billerica, MA) according to the manufacturer's instructions. AGO2 antibody was used for RIP (Cell Signaling Technology, Beverly, MA). Co-precipitated RNA was detected by qRT-PCR.

**Western blot analysis**. Briefly, total protein of LUSC tissue samples and cell lines was extracted using protein extraction reagent (Thermo Scientific) with a cocktail of proteinase inhibitors (Roche Applied Science, Switzerland) and a cocktail of phosphatase inhibitors (Roche Applied Science) according to its protocol. Equal amount of total protein (20 µg) was separated by 10% SDS–PAGE and transferred onto a PVDF membrane. After blocking for nonspecific binding, the membranes were incubated with antibody FOXM1 (1:3000 dilution; proteintech; 13147-1-AP), CENPA (1:1000 dilution; Abcam; ab45694), CENPB (1:1000 dilution; Abcam; ab25734), CCNB1 (1:1000 dilution, Cell Signaling Technology; #4134), or β-actin (1:10000 dilution; Sigma; A2228) overnight at 4 °C and followed by an incubation period of 1 h at room temperature with secondary antibody (1:4000, Bioword; 20330016-1). Bands were detected by a Bio-rad ChemiDoc XRS system. Full scans of the western blots shown in Figs. 4e, f, 5g and 6a, b and Supplementary Figs. 5c and 6c are provided in Source Data file.

**Statistics**. Results are presented as mean ± standard deviation of the mean. Statistical analyses were performed using Prism software (GraphPad Software), and consisted of analysis of variance followed by Student's *t*-test when comparing two experimental groups. A probability of 0.05 or less was considered statistically significant.

**Reporting summary**. Further information on research design is available in the Nature Research Reporting Summary linked to this article.

## Data availability

This microarray data are deposited in the NCBI Gene Expression Omnibus (GEO) datasets under the accession number GSE126533 (www.ncbi.nlm.nih.gov/geo). The authors declare that all the data supporting the findings in this study are available in this study and Supplementary Information. The source data underlying Figs. 3g, h, 4e, f, 5g and 6a, b and Supplementary Figs. 5c and 6c are provided as a Source Data file.

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

## Acknowledgements

This work was supported by grants from National Key Basic Research Program of China (973 Program: 2015CB553905), National Natural Science Foundation of China (81402278, 81472175, 81772461, 81702838), Shanghai Natural Science Foundation of China (16ZR1434700), Shanghai Municipal Commission of Health and Family Planning (201640055, 201640007, 20134007), Medical Guidance Project of Shanghai Science and Technology Committee (15411961500), and State Key Laboratory of Oncogenes and Related Gene (91-17-12).

## Author contributions

W.Q. supported and supervised the study. L.J. collected the LUSC tissues and supervised the study. Z.C., C.Y. and S.C. performed all the experimental validation assays and wrote the manuscript. H.W., H.J. and C.W. analyzed data. B.L., M.Q., C.Y., J.H., Q.Z., S.W., J.L. and W.Y. conducted the study. Y.L. provided study materials. F.Z. and M.Y. performed the animal experiments.

## Additional information

**Competing interests:** The authors declare no competing interests.

