## [Peer Review File · Nature Communications]

Reviewers' comments:

Reviewer #1, Expertise: noncoding RNA (Remarks to the Author):

Cheng et al. describe a new role for a circular RNA derived from TP63 in lung squamous cell carcinoma (LUSC). circTP63 is overexpressed in many tumors and its upregulation is correlated with larger tumor size and increased cell proliferation. As circTP63 shares miRNA binding sites with FOXM1 (particularly miR-873), the authors argue that circTP63 functions as a ceRNA and prevents miR-873 from decreasing the level of FOXM1. The manuscript is generally clear and will be of interest to cancer and circRNA researchers, although a number of points need to be clarified and strengthened with additional control experiments.

(1) Fig 1D: Given that 5 out of 6 circRNAs were not resistant to RNase R, there must be many false positives identified by the microarrays. This point needs to be made clear so that a reader does not over-interpret Fig 1A-B. Besides circTP63, are there any other circRNAs that the authors were able to validate? If not, the data in Fig 1A-B are of unclear significance.

(2) Fig 1E: Do the authors have any insights into why one of the tumors (S12) behaves differently than the 4 others?

(3) Fig 2: Northern blots should be performed to confirm that a circRNA of the expected size can be detected. This result would eliminate the possibility of trans-splicing yielding the back splice junction.

(4) Fig 3: What is the effect of knocking down TP63 mRNA on cell physiology? Does it cause any of the same phenotypes (e.g. proliferation changes) as the siRNA that knocks down the circRNA? The authors need to more clearly demonstrate that the phenotypes are definitely due to the circRNA and not due to subtle changes in TP63 mRNA expression. A siRNA to an exon not present in the circRNA would clarify this point.

(5) Fig 3: To truly prove that the circRNA is responsible for the phenotypes observed upon over-expression or siRNA knockdown, the authors should attempt to rescue the phenotypes. For example, upon treating cells with a siRNA to the back splice junction, can a lentivirus expressing a circRNA with a mutant back splice junction (such that it can not be targeted by the siRNA) rescue the proliferation phenotype? Such data would greatly strengthen the manuscript.

(6) Fig S4B: When describing these results, the main text is oversimplified. There is a significant change in BRCA1 levels in one of the cell lines. The authors instead write that levels "were not affected by circTP63" which is not an accurate statement.

(7) What is the cellular phenotype observed when circTP63 with a mutated miR-873 site is expressed from a lentivirus? The authors' model predicts that it should not affect proliferation, but is this the case?

(8) How does the expression of circTP63 compare to FOXM1 mRNA? In order for the authors' ceRNA hypothesis to be correct, circTP63 should be expressed much higher than FOXM1 mRNA.

Minor points

(1) Page 3, Line 17: "circRNAs as a new class of noncoding RNAs" – Given that some circRNAs may be translated, it is more appropriate to call these transcripts "regulatory RNAs."

(2) Page 6: It may be helpful to state in the main text how many circRNAs and mRNAs were profiled. This information is currently only in the methods.

(3) Fig 1F: What cutoff was used to say expression of the circRNA was significantly higher?

(4) Fig 2A: The exon with the nucleotide sequence above it is currently drawn in a confusing manner. It looks like the GTGAG part of the "GTGAGGGGC" label is in the intron when it is actually in the exon (a similar comment could be made for the label at the 3' end of the exon).

(5) Fig S3B: Please explain in more detail how the lentiviral vector produces a circRNA. Are complementary sequences present in the flanking introns?

(6) Fig S4A: It should be noted in the main text that there is a pretty good correlation between circTP63 and KIF18B levels.

Reviewer #2, Expertise: lung cancer (Remarks to the Author):

Here Cheng et al have identified a novel role for the circular RNAs, circTP63, in the progression of lung squamous cell carcinoma (LUSC). By analyzing the co-expression network of differentially expressed circRNAs and mRNAs in 5 paired human LUSC specimens, the authors found that circTP63 was frequently upregulated in LUSC tissues and correlated with tumor size and disease stage. Through in vitro and in vivo assays, they demonstrated that circTP63 promoted cell proliferation. They further found that circTP63 acted as a ceRNA to competitively bind to miR-873-3p and relieved the repression of miR-873-3p on FOXM1, thereby upregulating FOXM1 signaling and promoting the progression of LUSC. Overall, the study is interesting and novel. Some of my comments are listed below.

Major points:

1. What's the mechanism responsible for circTP63 upregulation in LUSC, especially in the link to the high expression of TP63 in LUSC? What's the expression pattern of circTP63 in other subtypes of lung cancer, e.g., ADC?
2. If the proliferation-promoting phenotype induced by circTP63 is due to its effects on FOXM1, why circTP63-induced FOXM1 upregulation had no effect on cell migration and invasion, given the facts that FOXM1 is identified as a major regulator of tumor metastasis including lung cancer (ref.39 and ref.44)? Does circTP63 expression have any correlation with the prognosis of LUSC patients?
3. What's the sequence of si-circTP63? Which region of circTP63 is targeted by this siRNA? To alleviate the concern about off-target effect, the results should be confirmed with a second siRNA.
4. As the effect of circTP63 knockdown on cell proliferation of SW900 is very marginal (Fig 3C), the impact of circTP63 knockdown upon tumor growth should be demonstrated in vivo. In addition, IHC staining for PCNA or Ki67 in xenograft tumors should be provided and the level of circTP63 should also be shown.
5. It would be nice to see if the effect of si-circTP63 on cell cycle and proliferation could be reversed by FOXM1 overexpression.
6. The authors showed that CENPA and CENPB were regulated by circTP63 through FOXM1, whereas they did not evaluate the contribution of CENPA and CENPB to circTP63 action. To draw the conclusion that "CENPA and CENPB as the downstream targets of FOXM1 are involved in the process of circTP63 promoting LUSC cell proliferation", it is important to see if knockdown these proteins could mimic the phenotypes of siFOXM1 in circTP63-expressing cells.
7. Since authors reveal a critical role of miR-873-3p-FOXM1-CENPA/B pathway in the circTP63-mediated proliferation of LUSC cells, and authors have available samples from the in vivo experiment in a mouse xenograft model, as well as the clinical samples, it would be interesting to observe whether this pathway is altered in correlation with circTP63 expression in these tissue samples.
8. The clinical information of tissue samples used for microarray as well as for validation in the study should be described in more detail in the Methods part.

Minor points:

1. In Fig 2C, how to explain the increase of circTP63 at 4hrs?
2. In bottom panel of Fig 5F, what is the meaning of comparison between the enrichment of circTP63 and FOXM1 in circTP63-overexpressed cells?
3. In Fig S3A, what's the expression pattern of circTP63 in human normal lung cell line?
4. The manuscript at this stage needs extensive correction of syntax, spelling, and grammatical errors. For example, on page 2 line 8, "upreglation" should be "upregulation"; on page 3 line 3, "newlydiagnosed" should be "newly diagnosed"; on page 6 line 9, "Figure 1aand" should be "Figure 1a and", on page 11 line 19, "Figure 6SA" should be "Figure S6A".

Reviewer #3, Expertise: Forkhead proteins (Remarks to the Author):

Cheng et al. describe circTP63 involvement in lung carcinoma development. They combine expression analysis of paired samples with bioinformatics to identify circTP63 as overexpressed in lung carcinoma cell lines. They then perform experiments showing that differential circTP63 expression affects proliferation and changes cell cycle distribution. They identify FOXM1 as possible mediator of circTP63 and show that this is likely through circTP63 acting as a sponge for miR-873-3p.

Taken together most conclusions drawn by the authors are supported by the data presented.

General comment

Figure 1 shows that over 7000 circRNAs and almost 3000 mRNAs show differential expression between paired samples. I think that from that background it is naïve to come with an extreme reductionist approach to show that 1 circRNA, 1 miR and 1 target explains it all. To me this shows that either they are just scratching the tip of the iceberg or that one has to conclude that the individual relevance (for cancer) of all these differential expressions is marginal.

Unfortunately, I tend to consider the latter conclusion to be more likely.

circTP63 is one of 7000 and can act as a sponge for 20+ miRNAs (fig 5) albeit with varying efficiency. Each miRNA will have hundreds if not more target mRNAs (depending on what prediction database one uses). Mathematics would then predict that the false discovery rate is as such that one can claim regulation by circTP63 of any gene product without being wrong.

In that perspective it is almost odd that only a subset of the known FOXM1 target genes is deregulated by circTP63 if FOXM1 is the target. So, in conclusion. Although the experiments are performed well, I have a hard time understanding the relevance of the reported experiments in the context of all that is additionally deregulated in lung cancer (high mutation load).

Being so I think that the manuscript represents little novelty. circRNAs are already extensively implied in cancer and gene regulation by acting as a sponge is also well documented.

Specific remarks

(-) I used target scan for miR-873-3p prediction and actually FOXM1 is predicted with really low score with 1 not well conserved seed site.

So what about all the other targets that score better in this prediction algorithm ?. This illustrates the highly selective biased cherry picking approach of the authors.

(-) Same for FOXM1 target genes, how do they explain that not all FOXM1 targets are regulated ?

Point-by-Point Response to Referees' Comment

Reviewer #1, Expertise: noncoding RNA (Remarks to the Author):

Cheng et al. describe a new role for a circular RNA derived from TP63 in lung squamous cell carcinoma (LUSC). circTP63 is overexpressed in many tumors and its upregulation is correlated with larger tumor size and increased cell proliferation. As circTP63 shares miRNA binding sites with FOXM1 (particularly miR-873), the authors argue that circTP63 functions as a ceRNA and prevents miR-873 from decreasing the level of FOXM1. The manuscript is generally clear and will be of interest to cancer and circRNA researchers, although a number of points need to be clarified and strengthened with additional control experiments.

Major concerns

(1) Fig 1D: Given that 5 out of 6 circRNAs were not resistant to RNase R, there must be many false positives identified by the microarrays. This point needs to be made clear so that a reader does not over-interpret Fig 1A-B. Besides circTP63, are there any other circRNAs that the authors were able to validate? If not, the data in Fig 1A-B are of unclear significance.

Response: Thanks a lot for your good comments. Figure 1d showed that hsa_circ_0068515 (termed circTP63) was more resistant to RNase R treatment compared to TP63 linear isoform. Other 5 circRNAs had the following three situations: 1) hsa_circ_0026398 and hsa_circ_0074026 could not be detected due to their very low abundance in lung cancer cells, but they could successfully be amplified in the cDNA of HEK-293T cells and Huh7 (a hepatocellular carcinoma cell line) (see the following attached Figure a). Additionally, their back-spliced junction could be confirmed by Sanger sequencing (see the following attached Figure b). 2) For hsa_circ_0026443, we tried different primers to amplify this circRNA, but all of these primers caused nonspecific amplification. 3) The other two circRNAs hsa_circ_0026414 and hsa_circ_0019089 were sensitive to RNase R.

According to your advice, among of the top 20 dysregulated circRNAs we further validated other 10 circRNAs (including 5 upregulated and 5 down downregulated) by Sanger sequencing and qRT-PCR. Our results showed that all of them were consistent with the result of microarray.

We described these results in “Result” section of the revised manuscript (page 6, line 21-23; and page 7, line 1-3) and added into Figure S1e-f as well as their legends.

(a) RT-PCR products with divergent primers in different cell lines (293T: human embryonic kidney cell line; Huh7: a hepatocellular carcinoma cell line; H2170 and H1703: lung squamous cell carcinoma cell lines). (b) Sanger sequencing for the back-spliced junction of candidate circRNAs

(2) Fig 1E: Do the authors have any insights into why one of the tumors (S12) behaves differently than the 4 others?

Response: Thank you for your question. Actually, we had noticed that the S12 tumor behaved differently when we overviewed the microarray data. The reason may be that S12 has lower stage (Stage IA) while others have higher stage with at least IIA (Supplementary Table 5). The S12 is at the early phase of tumor progression, and there may be less different in circTP63 expression between tumorous tissue and its matched adjacent nontumorous tissue. Another reason may be tumor heterogeneity in different patient.

(3) Fig 2: Northern blots should be performed to confirm that a circRNA of the expected size can be detected. This result would eliminate the possibility of trans-splicing yielding the back splice junction.

Response: Thanks a lot for your advice. According to your comments, we have performed northern blots to confirm circTP63 at 295nt with a probe targeted the back-spliced junction, and added into “Result” section of our revised manuscript (page7, line20-22) and Figure 2c as well as its legends.

(4) Fig 3: What is the effect of knocking down TP63 mRNA on cell physiology? Does it cause any of the same phenotypes (e.g. proliferation changes) as the siRNA that knocks down the circRNA? The authors need to more clearly demonstrate that the phenotypes are definitely due to the circRNA and not due to subtle changes in TP63 mRNA expression. A siRNA to an exon not present in the circRNA would clarify this point.

Response: Thank you so much for your advice. According to your comments, we knocked down the TP63 ($\Delta Np63$) by siTP63#1 and siTP63#2 (see the following attached Figure a), and found that knockdown of TP63 inhibited the growth of H1703 cells (see the following attached Figure b), suggesting TP63 has the similar effect of circTP63 on proliferation. In order to more clearly identify the phenotypes are due to the circTP63 instead of subtle changes in TP63 mRNA expression, siTP63#1 and siTP63#2 targeted TP63 but not in circTP63 were employed to reduce the proliferation of H1703, and simultaneously circTP63 was overexpressed to test whether it can rescue the role of siTP63#1 and siTP63#2. Results showed that overexpression of circTP63 could partially rescue the effect of siTP63, suggesting the phenotypes caused by circTP63 is independent on TP63 (see the following attached Figure c).

(a) Efficiency of siTP63 knockdown by siRNAs in H1703 cells. (b) Cell proliferation of H1703 cells with TP63 knocking down. (c) Proliferation inhibition of H1703 cells with TP63 knockdown could be rescued by circTP63. * $p < 0.05$; ** $p < 0.001$; *** $p < 0.001$.

(5) Fig 3: To truly prove that the circRNA is responsible for the phenotypes observed upon over-expression or siRNA knockdown, the authors should attempt to rescue the phenotypes. For example, upon treating cells with a siRNA to the back splice junction, can a lentivirus expressing a circRNA with a mutant back splice junction (such that it can not be targeted by the siRNA) rescue the proliferation phenotype? Such data would greatly strengthen the manuscript.

Response: We appreciate your great advice. We mutated the si-circTP63 targeted back splice junction by reversing 5' end of TP63 mRNA sequence "GCCAACA" into "ACAACCG" and 3' end of TP63 mRNA sequence "GTGAGGGGCCGT" into "TGCCGGGGAGTG". The circTP63-si-mut plasmids was constructed with the mutant back splice junction of "ACAACCGTGCCGGGGAGTG" (Sanger sequencing confirmed the mutant back-spliced junction; see the following attached Figure), and co-transfected with si-circTP63 into H1703 cells to determine whether it can affect the cell proliferation. Results showed that circTP63-si-mut rescued the proliferation phenotype and promoted

cell growth (see Figure S3e). We added the result into our revised manuscript (page 9, line 3-11) and Figure S3e as well as its legends.

Sanger sequencing for the mutant back-spliced junction of circTP63

(6) Fig S4B: When describing these results, the main text is oversimplified. There is a significant change in BRCA1 levels in one of the cell lines. The authors instead write that levels “were not affected by circTP63” which is not an accurate statement.

Response: According to your advice, we replaced the sentence “However, mRNA levels of KIF18B and BRCA1 were not affected by circTP63” by “For BRCA1, although there was a significant change in H2170 cells with circTP63 overexpression (Figure S4b), no significant correlation between the expression of circTP63 and BRCA1 in LUSC tissues was observed (Figure S4a)” in the “Result” section of our revised manuscript (page 11, line 3-6).

(7) What is the cellular phenotype observed when circTP63 with a mutated miR-873 site is expressed from a lentivirus? The authors’ model predicts that it should not affect proliferation, but is this the case?

Response: Thank you for your good question. Yes, circTP63 with a mutated miR-873-3p site didn’t affect cell proliferation. We mutated miR-873-3p binding site on circTP63, and transfected the circTP63-miR-mut into H226 and H2170 cells. Results showed there was no significant difference in proliferation compared with the group transfected with vector (see Figure S5f). We added the result into our revised manuscript (page 14, line 2-6) and Figure S5f as well as its legends.

(8) How does the expression of circTP63 compare to FOXM1 mRNA? In order for the authors' ceRNA hypothesis to be correct, circTP63 should be expressed much higher than FOXM1 mRNA.

Response: Thank you for your good question. The expression of FOXM1 mRNA was higher than the expression of circTP63 in almost all LUCS cell lines (see the following attached Figure). Similar phenomenon was also observed in other circRNAs. For example, circMTO1 was considered as the sponge of miR-9 to upregulate p21 expression and suppress hepatocellular carcinoma (HCC) progression. The relative expression of circMTO1 was 0.00001 while the relative expression of p21 was about 0.004 in HCC cell line SMMC-7721 (showed in Figure 5A and B of this paper ^[1]). Although the expression of circMTO1 was lower than its target p21, it is still possible to play its role through ceRNA mechanism. In general, circRNAs are expressed at low levels ^[2], which mean most circRNAs may have lower abundance than their target. But circRNAs are stable and have a long half-life, which help them to work longer as functional molecules in cell. This may explain circRNAs with lower expression can act as ceRNAs to regulate target molecules with the higher expression.

qRT-PCR analysis for expression levels of circTP63, miR-873-3p and FOXM1 in LUSC cell lines

[1] Han, D. et al. Circular RNA circMTO1 acts as the sponge of microRNA-9 to suppress hepatocellular carcinoma progression. *Hepatology*66, 1151-1164 (2017).

[2] Xiang, L. et al. The Biogenesis, Functions, and Challenges of Circular RNAs. *Molecular cell* 71(3):428-442(2018).

Minor points

(1) Page 3, Line 17: “circRNAs as a new class of noncoding RNAs” – Given that some circRNAs may be translated, it is more appropriate to call these transcripts “regulatory RNAs.”

Response: Thank you for your advice. We revised this sentence as “circRNAs as a new type of regulatory RNAs” in our revised manuscript (page3, line16-17).

(2) Page 6: It may be helpful to state in the main text how many circRNAs and mRNAs were profiled. This information is currently only in the methods.

Response: We added the information of microarray chip in the main text of our revised manuscript (page 6, line 4-6).

(3) Fig 1F: What cutoff was used to say expression of the circRNA was significantly higher?

Response: The $\text{Log}_2(\text{T/N expression})$ values were used to analyze the expression level of circTP63 in the LUSC samples (n=35). We identified $\text{Log}_2(\text{T/N expression})$ value >1 as significantly higher expression, which < -1 as lower expression, and between -1 and 1 as no significant change. We added the histogram of circTP63 expression in 35 LUSC tissues into Figure1f and legend as well.

(4) Fig 2A: The exon with the nucleotide sequence above it is currently drawn in a confusing manner. It looks like the GTGAG part of the “GTGAGGGGC” label is in the intron when it is actually in the exon (a similar comment could be made for the label at the 3' end of the exon).

Response: Thank you so much for your kind advice. We corrected the Figure 2a by moving the “GTGAG” to the 5’end of the exon part and the “CCAACA” to the 3’end of the exon part.

(5) Fig S3B: Please explain in more detail how the lentiviral vector produces a circRNA. Are complementary sequences present in the flanking introns?

Response: Circular RNA expression vector PLCDH is commercially available (GS0104, Guangzhou Genesee Biotech Co, China). To induce circTP63 circularization in cell, side flanking repeat sequences were added to both ends of the 295nt sequences (circTP63). The front circular frame contains the endogenous flanking genomic sequences with EcoRI restriction enzyme site and an AG splice acceptor, and the back circular frame contains part of the inverted upstream sequence with BamHI site and a GT splice donor. The flanking introns were complementary, and the sequences were listed in Supplementary Table 6.

(6) Fig S4A: It should be noted in the main text that there is a pretty good correlation between circTP63 and KIF18B levels.

Response: We really appreciate your advice. Figure S4a showed there is a correlation between circTP63 and KIF18B levels. However, when circTP63 was knocked down in H1703 cell or overexpressed in H2170 cells, no significant change of KIF18B happened when circTP63 was knocked down in H1703 cell or overexpressed in H2170. According to you advice, we amended the description of circTP63 and KIF18B in our main text (page 11, line 1-3)

Reviewer #2, Expertise: lung cancer (Remarks to the Author):

Here Cheng et al have identified a novel role for the circular RNAs, circTP63, in the progression of lung squamous cell carcinoma (LUSC). By analyzing the co-expression network of differentially expressed circRNAs and mRNAs in 5 paired human LUSC specimens, the authors found that circTP63 was frequently upregulated in LUSC tissues

and correlated with tumor size and disease stage. Through in vitro and in vivo assays, they demonstrated that circTP63 promoted cell proliferation. They further found that circTP63 acted as a ceRNA to competitively bind to miR-873-3p and relieved the repression of miR-873-3p on FOXM1, thereby upregulating FOXM1 signaling and promoting the progression of LUSC. Overall, the study is interesting and novel. Some of my comments are listed below.

Major points:

1. What's the mechanism responsible for circTP63 upregulation in LUSC, especially in the link to the high expression of TP63 in LUSC? What's the expression pattern of circTP63 in other subtypes of lung cancer, e.g., ADC?

Response: Thank you so much for your comments. The upregulation of parental gene or promotion of back-splicing may contribute to the mechanism for circTP63 upregulation in LUSC. According to your comments, we examined TP63 expression in the 35 paired samples of LUSC and found the expression of TP63 in LUSC tissues was significantly higher compared to corresponding adjacent nontumorous tissues (see the following attached Figure a). circTP63 expression was positively correlated with TP63 expression ($r=0.719$, $p<0.0001$) (see the following attached Figure b), suggesting that high expression of circTP63 is associated with higher expression of TP63. The back-splicing of circRNAs can be regulated by RNA-binding proteins ^[1]. It has been reported that DHX9, ADAR1 and QKI can broadly regulate the biogenesis of circRNAs ^[2-4]. By Oncomine data-mining analysis (<https://www.oncomine.org/>), we found that DHX9 and QKI were upregulated in LUSC tissues as compared to lung tissues (see the following attached Figure c and d), suggesting these proteins may be related to the upregulation of circTP63, but further studies need to be explored.

(a) qRT-PCR analysis for the expression of TP63 in the 35 paired samples of LUSC. (b) Correlation analysis of circTP63 and TP63 mRNA. (c-d) OncoPrint data-mining analysis for the expression of DHX9 (c) and QKI (d) in LUSC

According to your advice, we analyzed expression level of circTP63 in lung adenocarcinoma (LUAD). Interestingly, we found that circTP63 was significantly downregulated in 32 LUAD tissues compared to their corresponding adjacent nontumorous tissues (see the following attached Figure), and circTP63 expression in LUAD was opposite to what we observed in LUSC. This is a very interesting result, and significance of this difference needs to be further investigated.

qRT-PCR analysis for the expression of circTP63 in LUAD

- [1] Ashwal F R, et al. circRNA biogenesis competes with pre-mRNA splicing. *Molecular cell*56:55-66 (2014).
[2] Ivanov A, et al. Analysis of intron sequences reveals hallmarks of circular RNA biogenesis in animals. *Cell reports*10:170-177(2015).
[3] Aktas T, et al. DHX9 suppresses RNA processing defects originating from the Alu invasion of the human genome. *Nature* 544:115-119(2017).
[4] Conn S J, et al. The RNA binding protein quaking regulates formation of circRNAs. *Cell*160:1125-1134(2015).

2. If the proliferation-promoting phenotype induced by circTP63 is due to its effects on FOXM1, why circTP63-induced FOXM1 upregulation had no effect on cell migration and invasion, given the facts that FOXM1 is identified as a major regulator of tumor metastasis including lung cancer (ref.39 and ref.44)? Does circTP63 expression have any correlation with the prognosis of LUSC patients?

Response: This is a good comment. Currently, it has been described that FOXM1 as a major regulator of tumor metastasis is mainly involved in lung adenocarcinoma cells, such as PC-9 and H1299 (Figure 2 in ref^[5]), H1650 (Figure 4 and 5 in ref^[6]) and A549 (Figure 7 in ref^[7]). There is no report about the migration and invasion effects of FOXM1 in LUSC cell lines. Maybe due to the differences between cell lines, we didn't observe the significant effect of circTP63-induced FOXM1 on cell migration and invasion in the two lung squamous cell lines (H226 and H2170) used in our study. Molecular mechanism of this difference between LUAD cell lines and LUSC cell lines is unknown and needs to be explored in future.

Upregulation of circTP63 was significantly correlated with a larger tumor size and higher TNM stage in 35 LUSC patients (Table 1) and it would be correlation with the prognosis of LUSC patients. Due to limitation on sample size and follow-up time, we didn't show preliminary results of correlation analysis between circTP63 expression and prognosis of LUSC patients in this study (see the following attached Figure). The preliminary Kaplan-Meier analysis showed that patients with higher level of circTP63 were more likely to be a poor overall survival (OS), although p value was not significant ($p=0.2930$, see the following attached Figure).

Preliminary Kaplan-Meier analysis for the correlation between circTP63 expression and overall survival

[5] Xu, N. et al. FoxM1 is associated with poor prognosis of non-small cell lung cancer patients through promoting tumor metastasis. PloS one8, e59412 (2013).

[6] Peng, W. et al. FOXM1 Promotes Lung Adenocarcinoma Invasion and Metastasis by Upregulating SNAIL. Int J Biol Sci11(2): 186–198 (2015).

[7] Milewski D. et al. FOXM1 activates AGR2 and causes progression of lung adenomas into invasive mucinous adenocarcinomas. PLoS Genet13(12):e1007097 (2017).

3. What’s the sequence of si-circTP63? Which region of circTP63 is targeted by this siRNA? To alleviate the concern about off-target effect, the results should be confirmed with a second siRNA.

Response: The sequence of si-circTP63 is “GCCAACAGUGAGGGGCCGU” which is listed in Supplementary Table 6. This siRNA targets the back-splicing region of circTP63. “GCCAACA” belongs to the 3’ end of the linear TP63 (the part of 11 exon) while “GUGAGGGGCCGU” belongs to the 5’ end of the linear TP63 (the part of 10 exon). According to your advice, we designed a second siRNA (si-circTP63#2) to further exclude the off-target effect and confirm the function of circTP63. Results showed that knockdown of circTP63 by si-circTP63#2 had no effect on TP63 mRNA and inhibited the proliferation of H1703 cells *in vitro*, which is consistent with the effect of si-circTP63. We added these results in our revised manuscript (page 8, line14-15, 17, 19; and page 9, line1) and Figure S3c-d as well as figure legends.

4. As the effect of circTP63 knockdown on cell proliferation of SW900 is very marginal (Fig 3C), the impact of circTP63 knockdown upon tumor growth should be demonstrated *in vivo*. In addition, IHC staining for PCNA or Ki67 in xenograft tumors should be provided and the level of circTP63 should also be shown.

Response: Thank you so much for your good advice. According to your suggestion, we conducted tumor growth assay of circTP63 knockdown *in vivo*. 2×10^6 H1703 cells were subcutaneously injected into a single flank of each mouse (12 mice in total). Two weeks later, mice with palpable tumors (approximately 62 mm³) were randomly divided into two groups (6 mice per group), 50 nmol cholesterol-conjugated si-NC or si-circTP63 was intratumoral injected in the two groups three times per week for two weeks. Tumor growth was examined every 4-5 days. After mice were sacrificed, tumors were weighed and processed for further histological analysis. As results shown in Figure 3h, treatment of si-circTP63 significantly inhibited growth of H1703 *in vivo*. Immunohistochemistry revealed lower expression of Ki67 and PCNA in xenograft tumors of H1703 cells with circTP63 knockdown compared to si-NC group (see Figure S3i). The expression of circTP63 was decreased in si-circTP63 xenograft tumors (see Figure S3j). We added these descriptions in “Result” section (page 9, line 22; and page 10, line 1-8) and “Methods” section (page 24, line 3-9) of our revised manuscript.

5. It would be nice to see if the effect of si-circTP63 on cell cycle and proliferation could be reversed by FOXM1 overexpression.

Response: Thanks a lot for your good comment. According to your advice, FOXM1 was overexpressed in H1703 cells with circTP63 knockdown, and cell cycle and proliferation assays were performed. Results showed that FOXM1 overexpression significantly rescued the proliferation of H1703 (see Figure 4h) and promoted the cell cycle progression from G1/S to G2/M phase (see Figure S4f). The results were described in the “Results” section of our revised manuscript (page 11, line 19-22).

6. The authors showed that CENPA and CENPB were regulated by circTP63 through FOXM1, whereas they did not evaluate the contribution of CENPA and CENPB to circTP63 action. To draw the conclusion that “CENPA and CENPB as the downstream targets of FOXM1 are involved in the process of circTP63 promoting LUSC cell proliferation”, it is important to see if knockdown these proteins could mimic the phenotypes of siFOXM1 in circTP63-expressing cells.

Response: Thank you so much for your valuable advice. According to your suggestion, we knocked down CENPA and CENPB, respectively, in H226 and H2170 cells with circTP63 overexpression. Our results showed that CENPA or CENPB knocked down alone diminished the effect of circTP63 overexpression on H226 or H2170 proliferation (see Figure S6a-b). When CENPA and CENPB were knocked down together, more remarkable suppression effect on cell proliferation (see Figure 6c) was observed. Results showed that knockdown of CENPA and CENPB could mimic the phenotypes of siFOXM1 in the LUSC cell lines with circTP63 overexpression. These results were described in the “Results” section of revised manuscript (page 14, line 18-22; and page 15, line 1-3).

7. Since authors reveal a critical role of miR-873-3p-FOXM1-CENPA/B pathway in the circTP63-mediated proliferation of LUSC cells, and authors have available samples from the in vivo experiment in a mouse xenograft model, as well as the clinical samples, it would be interesting to observe whether this pathway is altered in correlation with circTP63 expression in these tissue samples.

Response: Thank you for your good comments. According to your advice, we analyzed the expression of FOXM1, CENPA and CENPB in xenograft tumors by qRT-PCR. Result showed that lower expression of FOXM1, CENPA, and CENPB in si-circTP63 group than si-NC group (see Figure S6c). In addition, we detected the expression of CENPA and CENPB in the 35 paired samples of LUSC by qRT-PCR and analyzed the correlation between circTP63 with CENPA or CENPB. Results showed that the expression of CENPA and CENPB, particularly CENPA, was remarkably increased in

LUSC tissues as compared to their corresponding adjacent nontumorous tissues (see Figure S6d), and was positively related to the expression of circTP63 (see Figure S6e). These data indicate that circTP63 promotes the cell proliferation through the FOXM1-CENPA/B pathway. We added these results in the “Result” section of our revised manuscript (page15, line 4-13).

8. The clinical information of tissue samples used for microarray as well as for validation in the study should be described in more detail in the Methods part.

Response: Thank you for your advice. According to your suggestion, we added more detail description about the tissue samples used for both microarray analysis and validation in the “Methods” part (page 20, line 3-10) and revised Supplementary Table 5.

Minor points:

1. In Fig 2C, how to explain the increase of circTP63 at 4hrs?

Response: Thank you for your good question. It has been described that circRNAs are increased after treatment of Actinomycin D, such like studies on circSMARC5^[8] and circHIPK3^[9]. Results in their papers showed the increase of circRNAs at 4hrs or 8hrs after treatment of Actinomycin D (Figure 2E of circSMARC5^[8]; Figure 3C of circHIPK3^[9]). We speculate that following reasons may be involved: 1) although the synthesis of pre-RNA is inhibited after treatment of Actinomycin D, there are still a number of pre-RNAs left and they can be spliced into circRNAs, and degradation of circRNAs is slower. 2) When cells face a stress (such as starved, UV radiation, Actinomycin D treatment, etc), RNA synthesis may be suddenly raised in the earlier phase of the stress.

[8] Yu, J. et al. Circular RNA cSMARCA5 inhibits growth and metastasis in hepatocellular carcinoma. *Journal of hepatology*68, 1214-1227 (2018).

[9] Qiu, Z. et al. Circular RNA profiling reveals an abundant circHIPK3 that regulates cell growth by sponging multiple miRNAs. *Nat Commun*7:11215 (2016).

2. In bottom panel of Fig 5F, what is the meaning of comparison between the enrichment of circTP63 and FOXM1 in circTP63-overexpressed cells?

Response: We are sorry for this mistake. We deleted this comparison in the revised manuscript

3. In Fig S3A, what's the expression pattern of circTP63 in human normal lung cell line?

Response: We tested the expression of circTP63 in BEAS-2B (human bronchial epithelial cells) and HFL-1 (human fetal lung fibroblast). Results showed that circTP63 has lower abundance in human normal lung cell lines than LUSC cell lines (see Figure S3a). We added the result into our revised manuscript (page 8, line 6-8) and complemented "method" section of our revised manuscript (page 20, line 14 and 17-18)

4. The manuscript at this stage needs extensive correction of syntax, spelling, and grammatical errors. For example, on page 2 line 8, "upreglation" should be "upregulation"; on page 3 line 3, "newlydiagnosed" should be "newly diagnosed"; on page 6 line 9, "Figure 1aand" should be "Figure 1a and", on page 11 line 19, "Figure 6SA" should be "Figure S6A".

Response: We are so sorry for these mistakes. We corrected the spelling and grammatical errors in our revised manuscript.

Reviewer 3: Expertise: Forkhead proteins (Remarks to the Author):

Cheng et al. describe circTP63 involvement in lung carcinoma development. They combine expression analysis of paired samples with bioinformatics to identify circTP63 as overexpressed in lung carcinoma cell lines. They then perform experiments showing that differential circTP63 expression affects proliferation and changes cell cycle distribution. They identify FOXM1 as possible mediator of circTP63 and show that this is likely through circTP63 acting as a sponge for miR-873-3p.

Taken together most conclusions drawn by the authors are supported by the data presented.

General comment

Figure 1 shows that over 7000 circRNAs and almost 3000 mRNAs show differential expression between paired samples. I think that from that background it is naïve to come with an extreme reductionist approach to show that 1 circRNA, 1 miR and 1 target explains it all. To me this shows that either they are just scratching the tip of the iceberg or that one has to conclude that the individual relevance (for cancer) of all these differential expressions is marginal.

Unfortunately, I tend to consider the latter conclusion to be more likely.

Response: Thanks a lot for your good comments. Cancer is a systemic and complicated disease, in which various alterations of genes and signaling pathways are involved. Scientists only try to identify one or several genes associated with the development and progression of cancer by high throughput screening methods in one paper. In this study, we focused on the study of circular RNA molecules related to LUSC. We first used a differential global gene expression analysis to investigate the expression profiles of circRNAs and their related mRNAs. Then, bioinformatics analysis was applied for screening and further functional study was performed. We found that cell cycle was the most enriched pathway in tumor tissues. So we focused on the differently expressed circRNAs involved in cell cycle. It has been reported in previous studies that this high throughput screening method is reasonable [1-3]. As circTP63, there may be other circRNAs involved in metastasis, metabolism, drug resistance, etc in LUSC. Our microarray data provide clues to further discovery other important circRNAs in LUSC for readers.

[1] Chen, K. et al. Methyltransferase SETD2-Mediated Methylation of STAT1 Is Critical for Interferon Antiviral Activity. *Cell* 170(3):492-506.e14 (2017).

[2] Song, Y. et al. Non-coding RNAs participate in the regulatory network of CLDN4 via ceRNA mediated miRNA evasion *Nat Commun* 8(1):289 (2107).

[3] Xie, L. et al. Deep RNA sequencing reveals the dynamic regulation of miRNA, lncRNAs, and mRNAs in osteosarcoma tumorigenesis and pulmonary metastasis. *Cell Death Dis* 9(7):772 (2018).

circTP63 is one of 7000 and can act as a sponge for 20+ miRNAs (fig 5) albeit with varying efficiency. Each miRNA will have hundreds if not more target mRNAs (depending on what prediction database one uses). Mathematics would then predict that the false discovery rate is as such that one can claim regulation by circTP63 of any gene product without being wrong.

Response: This is a good comment. It has been described that miRNA abundance is one of important limitations for ceRNA hypothesis [4]. Although lots of miRNAs and mRNAs are predicted to be regulated by circTP63, few miRNAs and mRNAs are the real targets. If miRNAs have low abundance, it is difficult that miRNAs regulate target mRNAs by binding with circRNAs. In this study, we analyzed the expression of miR-873-3p and circTP63 in LUSC cell lines and found that the abundance of miR-873-3p was comparable to that of circTP63 (Figure S5d). Based on the ceRNA hypothesis, we further performed luciferase reporter system to screen specific miRNAs involved in circTP63-miRNA-FOXM1 network, and found that miR-873-3p was the miRNA sponged by circTP63 (Figure 5b).

[4] Tay Y. et al. The multilayered complexity of ceRNA crosstalk and competition. *Nature*505(7483):344-52 (2014).

In that perspective it is almost odd that only a subset of the known FOXM1 target genes is deregulated by circTP63 if FOXM1 is the target. So, in conclusion. Although the experiments are performed well, I have a hard time understanding the relevance of the reported experiments in the context of all that is additionally deregulated in lung cancer (high mutation load).

Response: Thank you so much for your good question. The known targets of FOXM1 are involved in tumorigenesis, cell proliferation, self-renewal, etc. In this study, circTP63 mainly affected LUSC cell proliferation. So, among of known targets of FOXM1 we chose a subset of known FOXM1 target genes associated with cell proliferation. As you mentioned, there is a high tumor mutational burden (e.g., EGFR, ALK, ROS1, and BRAF V600E, etc) in lung cancer cells, and the correlation between mutations of lung cancer and roles of circTP63 is very interesting. circTP63 may have different molecular

mechanism in different kind of lung cancer. Further study for this question needs to be explored.

Being so I think that the manuscript represents little novelty. circRNAs are already extensively implied in cancer and gene regulation by acting as a sponge is also well documented.

Response: Thank you so much for your comments. As you mentioned, many reports have showed the role of circRNA as a ceRNA in different diseases in recent years. To our knowledge, this is the first report that systematically explores the expression, clinical significance, function and mechanisms of circRNA in LUSC.

Specific remarks

(-) I used target scan for miR-873-3p prediction and actually FOXM1 is predicted with really low score with 1 not well conserved seed site.

So what about all the other targets that score better in this prediction algorithm?. This illustrates the highly selective biased cherry picking approach of the authors.

Response: Thank you so much for your good question. As you mentioned, different databases may have different algorithms and get the different results. Therefore, we predicted the co-binding miRNAs of circTP63 and FOXM1 by miRanda miRNA target prediction tool, not only depending on the binding score of miRNA and mRNA, but also the binding score of miRNA and circRNA. We predicted the binding score of miR-873-3p and circTP63 at 154, and at 150 with FOXM1, which was the intermediate level among all predicted miRNAs. According to the miRanda prediction results, the highest predictive binding score of miRNA to FOXM1 was miR-211-3p at 179, but it bound weakly to cirTP63, and the luciferase activity was only reduced by 15% in our luciferase reporter assay. Similarly, the other miRNAs that score better in binding to FOXM1 also showed lower binding ability to circTP63 than miR-873-3p.

(-) Same for FOXM1 target genes, how do they explain that not all FOXM1 targets are regulated ?

Response: Thank you so much for your good question. As mentioned above, in this study, circTP63 mainly affected LUSC cell proliferation. Therefore, among of FOXM1 target genes, we chose 8 cell cycle-related candidates (AURKA, AURKB, CDC25B, CENPA, CENPB, CENPF, PLK1, and CCNB1) to investigate, and CENPA and CENPB were identified. Regulation of genes and signal pathway are complicated and cross-talk. Those FOXM1 target genes are regulated not only by FOXM1, but also by other genes. For example, AURKA can be suppressed by TP53 [5]. PLK1 can be methylated by the methyltransferase SETD6 [6]. mTOR pathway can play roles in controlling the expression of CCNB1 and PLK1 through regulating histone lysine demethylase 4B (KDM4B) [7]. FOXM1 may not be the main or only regulator to those genes, so that not all of FOXM1 target genes were affected by circTP63 through targeting FOXM1. Further study for this question is needed.

[5] Dauch D. et al. A MYC-aurora kinase A protein complex represents an actionable drug target in p53-altered liver cancer. *Nat Med* (7):744-53 (2016).

[6] Feldman M. et al. The methyltransferase SETD6 regulates Mitotic progression through PLK1 methylation. *Proc Natl Acad Sci USA*116(4):1235-1240 (2019).

[7] Hui Ju Heish, et al. Systems biology approach reveals a link between mTORC1 and G2/M DNA damage checkpoint recovery. *Nat Commun*9: 3982 (2018).

Finally, we thank you and reviewers again for your important comments for revision.

Reviewers' comments:

Reviewer #1 (Remarks to the Author):

The authors have addressed most of my previous concerns, although there are still a few things that should be clarified as outlined below.

(1) Page 6, Lines 19-23: I commented in my previous review that it is very odd that most of the identified circular RNAs are not resistant to RNase R. The authors have added some additional qRT-PCR evidence to indicate that some backsplicing events may be real, but I think they need to clearly state in the manuscript that "some of the identified circRNAs may be false positives." By definition, circRNAs should not be degraded by an exonuclease like RNase R.

(2) Figure S5d: More details need to be provided in the methods as to how exactly the authors determined the copy numbers of circTP63 and miR-873-3p. In the figure, the y-axis says "relative expression level" but absolute expression levels need to be measured, not relative expression levels. Standard curves need to be used. This is a very important point as the circRNA needs to be expressed at high levels relative to the miRNA in order for it to have a functional consequence and be a ceRNA.

(3) Page 9, Line 17: The description of how exactly the lentivirus works is still unclear in the manuscript. I tried to figure out from the cloning primers how the WT and mutant versions of circTP63 were cloned, but it is not obvious. The lentiviral plasmid was obtained from a Chinese company, but their description is all in Chinese so I am unable to judge it. Where exactly are the EcoRI, BamHI, AG splice acceptor, and GT splice donor that the authors refer to in their response to reviewers? A clearer description (with a very clear figure) should be provided in the manuscript, especially to explain how the version with the mutant junction was made (Page 9, Lines 3-6).

(4) Figure S4b: Although the authors write that no significant change in KIF18B levels were observed with circRNA overexpression, the error bars are very small so I find it highly unlikely that the expression change is not significant.

(5) In the response to the reviewers (p.4), the authors provide data about the phenotypes when knocking down TP63 mRNA. These data need to be incorporated into the manuscript, not just the response to reviewers.

Minor Points:

(1) Page 7, Line 15: "bp" should be "nt"

(2) Figure 2C: A loading control is needed

Reviewer #2 (Remarks to the Author):

This revised manuscript is suitable for the publication except for some minor changes, e.g., including the correlation of circTP63 with prognosis, the correlation between circTP63 and TP63 in LUSC and ADC.

Reviewer #3 (Remarks to the Author):

The authors have commented upon my remarks and concerns about e.g. false discovery rates and although I do not agree with their response and sometimes get the impression that my remarks

are not fully understood, I do not see the added value of getting into a lengthy repetitive non-discussion. If the journal wishes these type of studies who am I to object.

Point-by-Point Response to Referees' Comment

We are truly grateful to the reviewers for providing constructive and thoughtful comments, which have helped us to significantly improve this manuscript. In response, we have revised the manuscript and figures to address the issues that were raised. We have incorporated some figures as you required in supplemental figures 2 and 3. Due to space limitation, the previous supplemental figures 3 had to be divided into two figures. As a result, the revised manuscript has seven supplemental figures.

Reviewer 1:

(1) Page 6, Lines 19-23: I commented in my previous review that it is very odd that most of the identified circular RNAs are not resistant to RNase R. The authors have added some additional qRT-PCR evidence to indicate that some backsplicing events may be real, but I think they need to clearly state in the manuscript that “some of the identified circRNAs may be false positives.” By definition, circRNAs should not be degraded by an exonuclease like RNase R.

Response: Thank you for your suggestion. At present, the majority of circRNAs from circRNA database, such as circBase, are defined as circRNA only because their back-splice junction could be captured by second-generation sequencing, therefore, there may be false positive. For example, the back-splice junction sometimes could be produced by trans-splicing of two or more mRNAs. So we totally agree with your comment.

Following your suggestion, we have now added the statement in Page 6, Lines 22-23 and Page 7, Lines 1-3 as “hsa_circ_0026398 and hsa_circ_0074026 could not be detected due to their very low abundance in lung cancer cells. For hsa_circ_0026443, we tried different primers to amplify this circRNA, but all of these primers caused

nonspecific amplification. hsa_circ_0026414 and hsa_circ_0019089 were sensitive to RNase R, suggesting that some identified circRNAs may be false positives”

(2) Figure S5d: More details need to be provided in the methods as to how exactly the authors determined the copy numbers of circTP63 and miR-873-3p. In the figure, the y-axis says “relative expression level” but absolute expression levels need to be measured, not relative expression levels. Standard curves need to be used. This is a very important point as the circRNA needs to be expressed at high levels relative to the miRNA in order for it to have a functional consequence and be a ceRNA.

Response: Thank you for your important suggestions. We used purified RT-PCR products to generate the standard curve for absolute quantification. Briefly, circTP63 and miR-873-3p form cDNAs were amplified, purified and measured. Then they were serially diluted to be as templates for qRT-PCR. We counted the copy numbers based on their concentration, then combined the different Ct values at different copy numbers to generate the standard curves (see the following attached Figure a and b). We have added the information in the “method” section of our revised manuscript (Page 23, Lines 9-15).

According to the standard curve, we calculated the copy numbers of circTP63 and miR-873-3p in six LUSC cell lines (see Figure S6d). Results showed that the absolute expression of circTP63 is much higher than miR-873-3p in most of tested LUSC cell lines, which means circTP63 is enough to sponge miR-873-3p for its function. We have replaced the previous Figure S5d with Figure S6d, and added the results in revised manuscript (Page13, Lines 19-22)

The standard curve of circTP63 (a) or miR-873-3p (b)

(3) Page 9, Line 17: The description of how exactly the lentivirus works is still unclear in the manuscript. I tried to figure out from the cloning primers how the WT and mutant versions of circTP63 were cloned, but it is not obvious. The lentiviral plasmid was obtained from a Chinese company, but their description is all in Chinese so I am unable to judge it. Where exactly are the EcoRI, BamHI, AG splice acceptor, and GT splice donor that the authors refer to in their response to reviewers? A clearer description (with a very clear figure) should be provided in the manuscript, especially to explain how the version with the mutant junction was made (Page 9, Lines 3-6).

Response: We have redrawn a clearer sketch map (See Figure S3b). The plasmid PLCDH-ciR was modified by adding front circular frame and back circular frame for promoting circularization. Between the circular frames, wildtype or junction mutant linear TP63 was cloned into the plasmid (see the left part of Figure S3b). The specific amplified fragments are described in the middle part of Figure S3b. EcoRI, BamHI, AG splice acceptor, and GT splice donor are all designed in the primers. Cyclization-mediated sequence- F/R are used for separating AG splice acceptor or GT splice donor from restriction enzyme sites to ensure the accuracy of back splicing. For the wildtype version of circTP63, the primer sequences coincide with linear TP63 (highlighted in yellow). The circTP63 siRNA targeted sequences are in orange text. As a result, the amplified fragment contains in this order: EcoRI site,

cyclization-mediated sequence-F, splice acceptor AG, linear TP63, splice donor GT, cyclization-mediated sequence-R, and BamHI site. For the mutant version of circTP63, the same method was performed except using different primers. We replaced “GCCAACA” (the sequence in the 5’ end of TP63 mRNA that is targeted by si-circTP63) with “ACAACCG” using a mutant primer-F. Similarly, we replaced “GTGAGGGGCCGT” (the sequence in the 3’ end of TP63 mRNA that is targeted by si-circTP63) with “TGCCGGGGAGTG” using mutant primer-R. When the recombinant plasmids are transfected into cells, the spliceosome recognizes the AG splice acceptor and GT splice donor. The linear TP63 or mutant linear TP63 is back spliced, and generated as circTP63 or circTP63-si-mut (see the right part of Figure S3b). We added the description into the “Method” part of our revised manuscript (Page 24, Lines 4-19).

This plasmid is also used to generate circSHPRH overexpression vector (Zhang M. et al., *Oncogene* 2018) and circAGFG1 overexpression vector (Yang R. et al., *Mol Cancer* 2019) by other researchers.

(4) Figure S4b: Although the authors write that no significant change in KIF18B levels were observed with circRNA overexpression, the error bars are very small so I find it highly unlikely that the expression change is not significant.

Response: Thank you for your comment. We now reanalyzed the data with unpaired T-test. The p value was 0.0253, which means significant. Although this result showed that circTP63 may affect the expression of KIF18B, the regulation is still much weaker than that on FOXM1. We have corrected the p value in Figure S5b and added the description in revised manuscript (Page 11, Lines 13-14).

(5) In the response to the reviewers (p.4), the authors provide data about the phenotypes when knocking down TP63 mRNA. These data need to be incorporated into the manuscript, not just the response to reviewers.

Response: Thank you for this suggestion. We have incorporated these data in revised manuscript (Page 9, Lines 20-22; and Page 10, Line 1) and Figure S3f and g

Minor Points:

(1) Page 7, Line 15: “bp” should be “nt”

Response: Thanks, we have now corrected (Page 8, Line 6).

(2) Figure 2C: A loading control is needed

Response: Thanks, we have added the total RNA as the loading control.

Reviewer #2:

This revised manuscript is suitable for the publication except for some minor changes, e.g., including the correlation of circTP63 with prognosis, the correlation between circTP63 and TP63 in LUSC and ADC.

Response: We appreciated your suggestion. We have now incorporated the correlation of circTP63 with prognosis in revised manuscript ((Page 7, Lines 20-22; and Page 8, Line 1) and Figure S2d. We have also added the correlation between circTP63 and TP63 in LUSC in revised manuscript (Page 7, Lines 13-18) and Figure S2b and c as well as their legends.

Reviewer #3:

The authors have commented upon my remarks and concerns about e.g. false discovery rates and although I do not agree with their response and sometimes get the impression that my remarks are not fully understood, I do not see the added value of

getting into a lengthy repetitive non-discussion. If the journal wishes these type of studies who am I to object.

Response: Thank you for reviewing our manuscript. Some papers are similar strategy and study design with our manuscript. For example, a paper published recently in Cell (Sujun Chen. et al., Cell 2019) identified 11.3% of highly abundant circRNAs which were essential for cell proliferation in prostate cancer by ultra-deep rRNA-depleted RNA sequencing, and showed that a circRNA molecule, circCSNKG3, promoted cell growth by interacting with miR-181.

Thank you again for your comments to our manuscript. Here, we would like to highlight our findings in this study. Firstly, this is the first report that systematically explores the expression, clinical significance, function and mechanisms of circRNA in LUSC. Secondly, a co-expression profiling of circRNA and mRNA in LUSC was first elucidated by global different expression microarray analysis. Thirdly, our data suggest that circTP63 is a valuable circRNA molecule and a promising potential biomarker in LUSC.

Finally, we would like to thank the reviewers again for their time and efforts to provide constructive remarks to our revised manuscript.

REVIEWERS' COMMENTS:

Reviewer #1 (Remarks to the Author):

I still have some concerns on the quality of the microarray data as most of the putative circRNAs are not resistant to RNase R. Regardless, it seems like the candidate that the authors focus on - circTP63 - is resistant to RNase R and the authors have done a number of experiments that support its functional significance in LUSC via miR-873-3p and FOXM1.

On p.11, Line 14: Please change "much weaker" to "weaker" to not exaggerate the data

Reviewer #2 (Remarks to the Author):

I have no more questions.

Point-by-Point Response to Referees' Comment

Reviewer 1:

(1) I still have some concerns on the quality of the microarray data as most of the putative circRNAs are not resistant to RNase R. Regardless, it seems like the candidate that the authors focus on - circTP63 - is resistant to RNase R and the authors have done a number of experiments that support its functional significance in LUSC via miR-873-3p and FOXM1.

Response : Thank you so much for your comments to the revised version of our manuscript.

(2) On p.11, Line 14: Please change "much weaker" to "weaker" to not exaggerate the data

Response: Thank you for your suggestion. We have corrected "much weaker" to "weaker" in the main text.

Finally, we thank reviewers again for your important comments for revision.